# Activation of subnanometric Pt on Cu-modified CeO₂ via redox-coupled atomic layer deposition for CO oxidation

Xiao Liu [1,2], Shuangfeng Jia [3], Ming Yang [4,6], Yuanting Tang[2], Yanwei Wen[2], Shengqi Chu [5], Jianbo Wang [3], Bin Shan [2✉] & Rong Chen [1✉]

Improving the low-temperature activity (below 100 °C) and noble-metal efficiency of automotive exhaust catalysts has been a continuous effort to eliminate cold-start emissions, yet great challenges remain. Here we report a strategy to activate the low-temperature performance of Pt catalysts on Cu-modified CeO₂ supports based on redox-coupled atomic layer deposition. The interfacial reducibility and structure of composite catalysts have been precisely tuned by oxide doping and accurate control of Pt size. Cu-modified CeO₂-supported Pt sub-nanoclusters demonstrate a remarkable performance with an onset of CO oxidation reactivity below room temperature, which is one order of magnitude more active than atomically-dispersed Pt catalysts. The Cu-O-Ce site with activated lattice oxygen anchors deposited Pt sub-nanoclusters, leading to a moderate CO adsorption strength at the interface that facilitates the low-temperature CO oxidation performance.

[1] State Key Laboratory of Digital Manufacturing Equipment and Technology, School of Mechanical Science and Engineering, Huazhong University of Science and Technology, 430074 Wuhan, Hubei, People's Republic of China. [2] State Key Laboratory of Materials Processing and Die and Mould Technology, School of Materials Science and Engineering, Huazhong University of Science and Technology, 430074 Wuhan, Hubei, People's Republic of China. [3] School of Physics and Technology, Center for Electron Microscopy, MOE Key Laboratory of Artificial Micro- and Nano-structures, and Institute for Advanced Studies, Wuhan University, 430072 Wuhan, Hubei, People's Republic of China. [4] General Motors Global Research and Development, Chemical Sciences and Materials Systems Lab, 3500 Mound Road, Warren, Michigan 48090, USA. [5] Institute of High Energy Physics, Chinese Academy of Sciences, 100049 Beijing, People's Republic of China. [6]Present address: Department of Chemical and Biomolecular Engineering, Clemson University, Clemson, South Carolina 29634, USA. ✉email: bshan@mail.hust.edu.cn; rongchen@mail.hust.edu.cn

The fuel economy of vehicles has been greatly improved with the development of turbocharged engines, exhaust heat recovery systems and hybrid powertrains[1,2]. Due to the much lower exhaust temperature in these vehicles, after-treatment catalysts must work efficiently at low temperatures for the recirculation usage of exhaust gas. The general consensus is that the next generation catalyst must be active in the sub 100 °C range to reduce or eliminate cold-start emissions[3–5]. To meet the requirement for >90% abatement of emissions (CO, HC, and $NO_x$) at 150 °C[6], the challenge is to develop highly efficient catalysts at low temperatures. Despite considerable progress made in low-temperature exhaust catalysts, such as Au nanoparticles and $Co_3O_4$[7,8], the harsh working environment of automotive catalysts would severely deteriorate the stability of such nanocatalysts. By far, Pt is still the most widely used catalyst component in the exhaust cleaning system due to its excellent reactivity and chemical stability[9].

Since the discovery of the strong metal–support interaction, there has been an explosion of investigations on reducible oxide-supported Pt catalysts to enhance the low-temperature activity, such as $Pt/CeO_2$, $Pt/FeO_x$, and $Pt/Co_3O_4$[10–14]. It has been demonstrated that the size of supported Pt catalysts is a key factor that affects the proportion of the interfacial active sites and interfacial interactions, directly correlating with the activity of Pt/oxide catalysts[15]. Pt nanoparticles with sizes around 1.6 nm on $CeO_2$ supports have been reported to exhibit a turnover frequency (TOF) of 0.2 s$^{-1}$ at 80 °C for CO oxidation, much greater than larger Pt nanoparticles with the same Pt mass loading[16–18]. To further increase the utilization rate of the Pt component, supported single atoms and sub-nanocluster catalysts have been extensively investigated due to their higher atomic efficiency[19–21]. While the atomically dispersed Pt atoms on oxide supports have been shown to exhibit higher activity than supported Pt nanoparticle catalysts in some reports[22,23], it is still under heated debate as a number of studies pointed out that Pt single atoms and sub-nanoclusters are merely bystanders for low-temperature reactions in similar catalytic systems[24–26]. Despite researcher efforts to promote $CeO_2$ supported Pt single atom catalysts using high temperature steam treatment, heteroatom doping and hydrogen thermal pretreatment, activities of such systems are nonetheless inferior to the activity of state-of-the-art $Pt/CeO_2$ catalysts with Pt nanocluster sizes in the range of 1.2–1.6 nm[5,13,16,27,28]. In order to further improve the atomic efficiency, the highly efficient Pt subnanometric catalysts supported on $CeO_2$ have been reported, which have shown enhanced low-temperature activity compared with atomically dispersed Pt catalysts[29,30]. It has also been reported that the catalytic activity of supported Pt single atoms and sub-nanoclusters is dependent on the dynamically evolving interfacial structures under reaction conditions[31–33]. At this juncture, a precise control of interfacial structures would have considerable potential in the development of highly efficient low-temperature catalysts. Recently, our group achieved such control of the interfacial structures of supported Pt nanoparticles and sub-nanoclusters by an atomic layer deposition (ALD) method[34,35]. The ALD method has also been successfully employed to prepare supported single atom catalysts by preventing adsorbed atoms on the supports from aggregating[36,37]. Nevertheless, since the deposition and diffusion of Pt atoms on oxide supports is sensitive to oxide surface structure and ALD protocols, a major challenge is the simultaneous control of the interfacial reducibility and structures for supported Pt catalysts, especially in the size range of sub-nanometer clusters to single atoms.

Here we report a strategy to precisely control interfacial structures of $CeO_2$ supported Pt catalysts by coupling oxide doping and Pt size control via a redox-coupled ALD method. The remarkable performance with an onset of CO oxidation reactivity below room temperature and turnover frequency (TOF) of 0.26 s$^{-1}$ at 80 °C is demonstrated for the Cu-modified $CeO_2$ supported Pt sub-nanoclusters. Both experimental and theoretical results reveal the finely constructed interface composed of Cu–O–Ce site and with deposited Pt sub-nanoclusters contributes to the bifunctional active site, with enhanced lattice oxygen activity and moderate CO adsorption strength activating the low-temperature catalytic performance.

## Results

**Structural characterizations of catalysts.** In our experiment we started from the $CeO_2$ nanorod supports with exposed controllable facets of (220) and (200) (see Supplementary Fig. 1 in Supplementary Information), which were prepared by a hydrothermal method as reported in our previous study[38]. As illustrated in Fig. 1a, Cu dopants were introduced into $CeO_2$ supports (denoted as $Ce_{0.99}Cu_{0.01}O_2$) to tune the surface reducibility and to anchor Pt precursors during the ALD process. The atomically dispersed Pt catalysts were deposited on two supports via regular Pt ALD with the sequence of $MeCpPtMe_3$-$O_2$, which were denoted as $Pt_1/Ce$ and $Pt_1/CeCu$. Notably, the Pt mass loading of $Pt_1/CeCu$ (1.51 wt%) is much larger than that of $Pt_1/Ce$ (0.63 wt%), due to the larger density of surface defects in Cu-doped samples for anchoring Pt precursors (Supplementary Fig. 2). The Cu K-edge X-ray absorption near edge structure (XANES) spectrum of $Pt_1/CeCu$ (Supplementary Fig. 3) shows that Cu dopants in $Cu^{2+}$ state are atomically dispersed in ceria support, in good agreement with previous studies[39]. Meanwhile, the absence of signals in the Cu 2p spectra indicates that no segregated $CuO_x$ species are formed in $CeO_2$ supports due to the low Cu concentration (0.20 wt% determined by ICP-OES). In order to fine tune the size of Pt, a redox-coupled ALD recipe of Pt with the sequence $MeCpPtMe_3$–$O_2$–$H_2$ was performed on two supports (denoted as $Pt_n/Ce$ and $Pt_n/CeCu$), utilizing the aggregation of Pt atoms under a reduced atmosphere[40].

The morphology of prepared Pt catalysts is characterized by aberration-corrected high-angle annular dark-field scanning transmission electron microscopy (HAADF-STEM). The atomically dispersed Pt atoms (white circles) are shown for $Pt_1/Ce$ in Fig. 1b, which is consistent with the 2097 cm$^{-1}$ peak detected in diffuse reflectance infrared Fourier transform spectroscopy (DRIFTS) spectrum of CO adsorption (Supplementary Fig. 4). Applying the same ALD process to $Pt_1/Ce$, the HAADF-STEM image of $Pt_1/CeCu$ also shows atomically dispersed Pt atoms (Fig. 1d). However, the peak at 2097 cm$^{-1}$ in the DRIFTS spectrum of CO adsorption on $Pt_1/CeCu$ is greater than that of $Pt_1/Ce$ (Supplementary Fig. 4), which can be attributed to the larger Pt mass loading of $Pt_1/CeCu$ induced by surface Cu dopants. The numbers of Pt atoms per unit area are also consistent with that calculated by the Pt mass loading of $Pt_1/Ce$ and $Pt_1/CeCu$ catalysts (Supplementary Fig. 5). Different from $Pt_1/Ce$ and $Pt_1/CeCu$, subnanometric Pt clusters are observed in the HAADF-STEM images of $Pt_n/Ce$ and $Pt_n/CeCu$ catalysts with the redox-coupled ALD (Fig. 1c, e), with mean particle sizes of 0.63 ± 0.18 and 0.75 ± 0.11 nm, respectively (Supplementary Fig. 5). The average distances of Pt clusters in a ceria nanorod are 4.94 ± 1.21 and 4.60 ± 1.51 nm for $Pt_n/Ce$ and $Pt_n/CeCu$, respectively (Supplementary Fig. 6). The inserted HAADF-STEM images also definitively show $CeO_2$ (220) facet-supported Pt clusters composed of aggregated Pt atoms. In addition, the DRIFTS spectrum of CO adsorption on $Pt_n/Ce$ and $Pt_n/CeCu$ shows a mix of linear- and bridged-bonded CO on Pt clusters with peaks at 2083, 2064, 1883, and 1838 cm$^{-1}$. No Pt diffraction signals were detected in the X-ray diffraction patterns

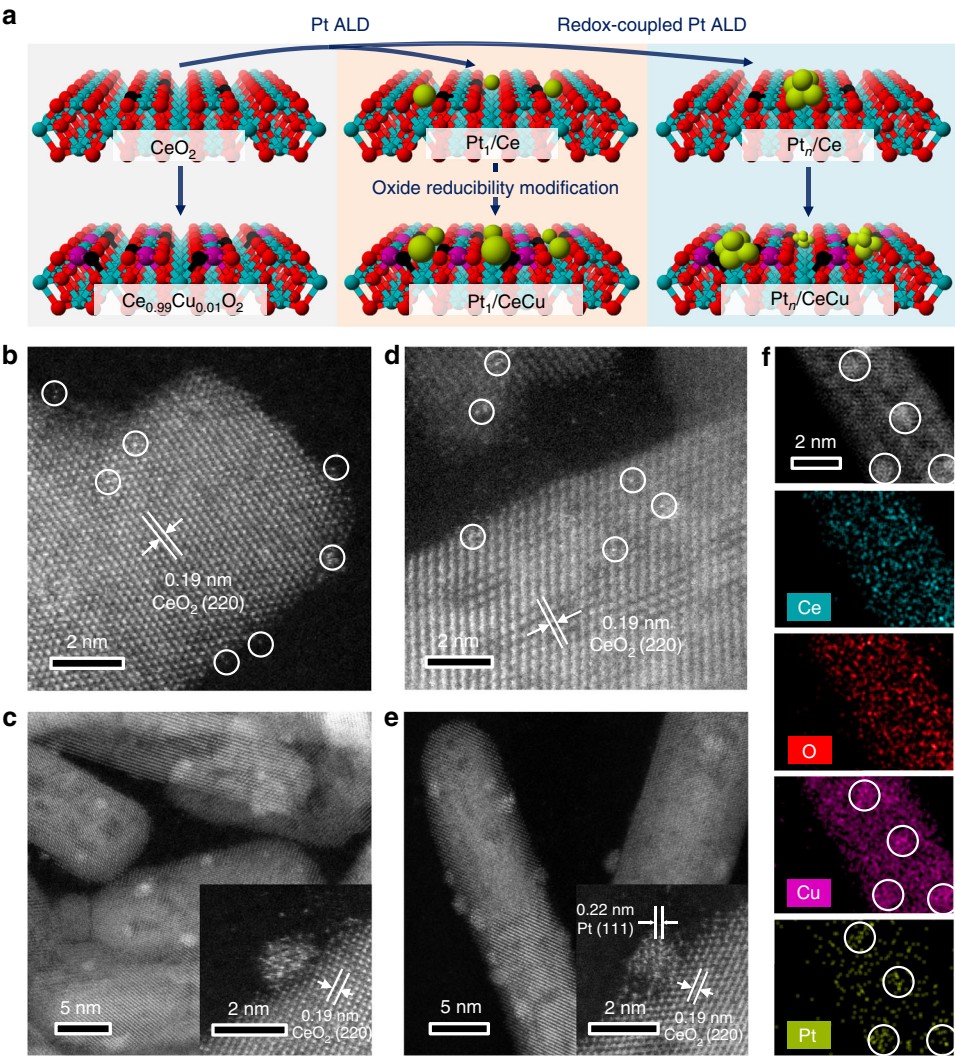

**Fig. 1 Preparation processes and electron microscopy characterizations of catalysts. a** Schematic illustration of preparation processes for supported Pt catalysts, with Ce, O, Pt, and Cu atoms shown as Red, Cyan, Green, and Purple, respectively. HAADF-STEM images of (**b**) $Pt_1$/Ce, (**c**) $Pt_n$/Ce, (**d**) $Pt_1$/CeCu, and (**e**) $Pt_n$/CeCu. Inserts in (**c**) and (**e**) show HAADF-STEM images of the $Pt_n$/Ce and $Pt_n$/CeCu interfaces. Atomically dispersed Pt atoms are marked in (**b**) and (**d**) by white circles. **f** EDS analysis of $Pt_n$/CeCu shows the locations of Ce, O, Cu, and Pt.

(Supplementary Fig. 7), which is due to the high dispersion and low crystallinity of Pt clusters. The presence of Cu in $Pt_n$/CeCu catalyst was further confirmed by energy dispersive spectroscopy (EDS) (Supplementary Fig. 8). STEM-EDS elemental maps (Fig. 1f) were produced to study the distribution of Cu in the catalyst. Clearly, the concentrations of Cu dopants near the Pt clusters are higher than other regions, which is beneficial to tune the reducibility of the support near the interfaces and affect the activity of composite catalyst.

**CO oxidation performance evaluation.** CO oxidation tests were then performed to investigate catalytic activities of supported Pt catalysts. Figure 2a shows the CO conversion rates of prepared catalysts as a function of reaction temperature. Interestingly, $Pt_n$/CeCu exhibits an onset of CO oxidation reactivity below the room temperature, and the $T_{50}$ (50% CO conversion temperature) is decreased to 34 °C, which is much lower than that of $Pt_n$/Ce (91 °C), $Pt_1$/CeCu (116 °C), and $Pt_1$/Ce (166 °C). The slightly shifts of CO conversion curves in cycling tests of $Pt_n$/CeCu (Supplementary Fig. 9) indicate the structural stability of the constructed Pt/oxide interface, which can be attributed to the gas-phase based

ALD method that has minimal detrimental effect on the surface structure of oxide supports[35]. The concentration of Cu dopants is kept low in $Ce_{0.99}Cu_{0.01}O_2$ to avoid the formation of $CuO_x$ species, and catalytic testing show that the contribution of our $Ce_{0.99}Cu_{0.01}O_2$ support to reactivity below 100 °C is negligible (Supplementary Fig. 10), in agreement with the comparable activity of Cu-doped $CeO_2$ catalyst in previous study[41]. It can be concluded that the enhanced activity of $Pt_n$/CeCu originates from the highly dispersed Pt sub-nanoclusters and the synergy of introduced Cu dopants in the $Ce_{0.99}Cu_{0.01}O_2$ supports. In order to eliminate the effects of Pt's mass loading for fair comparison, $Pt_n$/CeCu was diluted by an appropriate amount of $Ce_{0.99}Cu_{0.01}O_2$, and CO oxidation activity was determined. The catalyst still exhibits better low-temperature CO oxidation activity with a $T_{50}$ of 54 °C after dilution (Supplementary Fig. 11). In addition, the effect of Cu dopants concentration was also investigated (Supplementary Fig. 12). The CO oxidation activities go down when Ce/Cu molar ratio increases to 95:5, in part due to the separated $CuO_x$ species that adversely affect the interfacial structures of composite catalysts (Supplementary Fig. 13). Furthermore, the intrinsic activities of synthesized catalysts have been evaluated by kinetic tests as shown in Fig. 2b. The calculated apparent

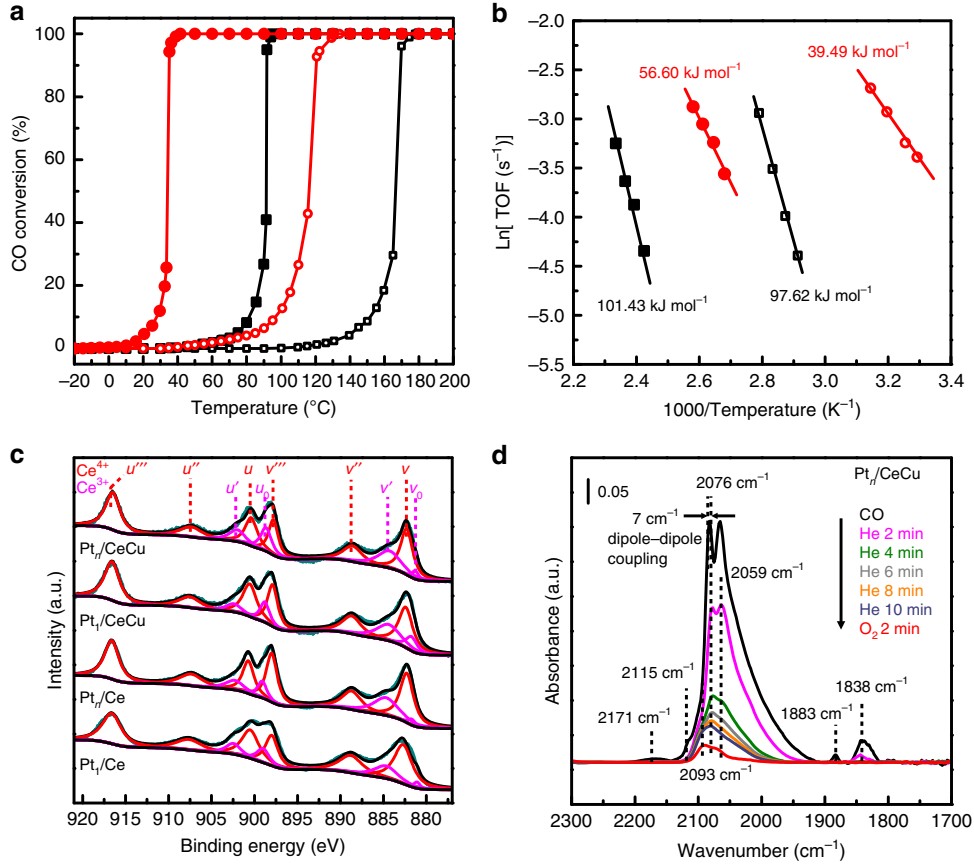

**Fig. 2 Catalytic properties and surface activities characterizations of catalysts. a** CO conversion and (**b**) reaction rates of catalysts as a function of reaction temperature. The catalysts are $Pt_1/Ce$ (black filled square), $Pt_n/Ce$ (black square), $Pt_1/CeCu$ (red filled circle) and $Pt_n/CeCu$ (red circle). **c** Ce 3$d$ XPS spectra of $Pt_1/Ce$, $Pt_n/Ce$, $Pt_1/CeCu$ and $Pt_n/CeCu$. **d** In situ DRIFTS spectra of CO adsorption and oxidation of $Pt_n/CeCu$. After CO exposure, He flow is continued and the spectra is recorded at 2, 4, 6, 8, 10 min. Subsequently, the flow is switched to 1% vol. $O_2$ balanced by $N_2$ and the spectrum is recorded at 2 min. Source data are provided as a Source Data file.

activation energy of $Pt_n/CeCu$ (39.49 kJ mol$^{-1}$) is much lower than that of $Pt_n/Ce$ (97.62 kJ mol$^{-1}$) and other two Pt single atoms catalysts. The reaction orders of CO and $O_2$ over $Pt_n/CeCu$ are close to zero (Supplementary Fig. 14), which indicates the negligible competitive adsorption between CO and $O_2$ during CO oxidation at the interfaces of $Pt_n/CeCu$. The activation energy of $Ce_{0.99}Cu_{0.01}O_2$ support is determined to be 77.60 kJ mol$^{-1}$ (Supplementary Fig. 15), close to that in previous studies[42–44]. The TOF of $Pt_n/CeCu$ catalyst reaches 0.26 s$^{-1}$ at 80 °C, which rivals other previously reported $Pt/CeO_2$ catalysts including Pt single atoms, clusters and nanoparticles (Supplementary Table 1).

**Interfacial structures and activities investigations**. The interfacial oxygen activation capability and CO adsorption strength were then determined to clarify the root cause of the superior low-temperature CO oxidation activity of $Pt_n/CeCu$. On one hand, the surface oxygen vacancies of catalysts were characterized by the concentrations of $Ce^{3+}$ ($[Ce^{3+}]$) calculated based on two sets of peaks in the Ce 3$d$ X-ray photoelectron spectroscopy (XPS) spectra as shown in Fig. 2c. The calculated $[Ce^{3+}]$ of catalysts followed the order: $Pt_n/CeCu > Pt_1/CeCu > Pt_n/Ce > Pt_1/Ce$, implying that Cu dopants can considerably increase the concentrations of surface oxygen vacancy, which is consistent with the O 1$s$ XPS results (Supplementary Fig. 16 and Supplementary Table 2). In addition, Raman spectrum analysis shows that $Pt_n/CeCu$ and $Pt_1/CeCu$ catalysts have a stronger defect-induced vibration mode than $Pt_n/Ce$ and $Pt_1/Ce$ (Supplementary

Fig. 17). The temperature-programmed reduction by hydrogen ($H_2$-TPR) has also been performed to investigate the surface reducibility of our prepared catalysts (Supplementary Fig. 18). The appearance of the reduction peak at 242 °C for $Ce_{0.99}Cu_{0.01}O_2$ support can be assigned to the reduction of Cu–O–Ce species[42,43]. Cu dopants also cause the shift of reduction peaks of surface oxygen (250–550 °C) and bulk oxygen (about 710 °C) of $CeO_2$ supports for $Pt_1/Ce$ and $Pt_n/Ce$ to low temperatures. The reducibility of surface oxygen follows the sequences of $Pt_n/CeCu > Pt_1/CeCu > Pt_n/Ce > Pt_1/Ce$, indicating that both Cu doping and change of Pt size can affect the surface reducibility of our prepared catalysts.

On the other hand, in situ DRIFTS spectra of $Pt_n/CeCu$ in Fig. 2d clearly show the loss of adsorbed CO molecules under He flow after CO exposure at the room temperature. The stretching signals of formed $CO_2$ molecules under CO flow also decrease, which are related to the decreased CO adsorption signals (Supplementary Fig. 19). When the flow is switched from He to $O_2$, the signal of adsorbed CO molecules is further weakened. The appearance of stretching bands of $CO_2$ molecule at 2360 cm$^{-1}$ and 2330 cm$^{-1}$ suggests that the adsorbed CO molecules on $Pt_n/CeCu$ can be further oxidized by activated $O_2$ molecules at the interface, which is typical in $Pt/CeO_2$ catalysts[45,46]. In order to eliminate the effects of interfacial active oxygen, the in situ DRIFTS spectra of CO adsorption of $Pt_n/CeCu$ after pretreatment under CO flow have been collected, which still show the loss of adsorbed CO molecules indicating weakened CO bonding strength on Pt clusters due to Cu dopants (Supplementary Fig. 20). As a

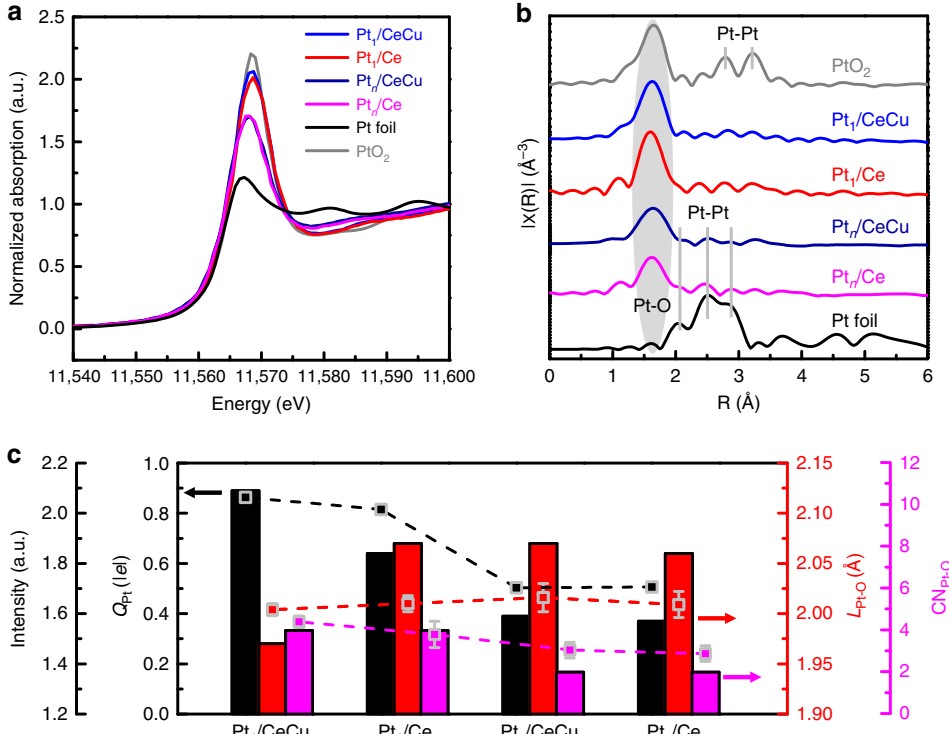

**Fig. 3 Interfacial structures characterizations of catalysts. a** Normalized Pt L$_{III}$-edge XANES and (**b**) $k^2$-weighted Fourier-transformed EXAFS spectra of Pt$_1$/CeCu, Pt$_1$/Ce, Pt$_n$/CeCu, Pt$_n$/Ce, Pt foil and PtO$_2$. **c** Q$_{Pt}$, calculated Bader charges of interfacial Pt atoms; L$_{Pt-O}$, bond lengths between Pt atoms and interfacial oxygen; and CN$_{Pt-O}$, coordinate number of interfacial Pt atoms, of Pt$_1$/CeCu, Pt$_1$/Ce, Pt$_n$/CeCu and Pt$_n$/Ce. The black square points are the white line intensity from XANES spectra. The red and magenta square points representing L$_{Pt-O}$ and CN$_{Pt-O}$ are the fits to the EXAFS spectra. Source data are provided as a Source Data file.

comparison, in situ DRIFTS spectra of Pt$_n$/Ce (Supplementary Fig. 21) show that Pt sub-nanoclusters are poisoned by strongly adsorbed CO molecules. The temperature-programmed desorption of CO has also been performed to investigate the desorption behavior of CO on Pt$_n$/CeCu and Pt$_n$/Ce (Supplementary Fig. 22). The CO desorption found at −19 °C for Pt$_n$/CeCu catalyst along with the formation of CO$_2$ can be attributed to the CO oxidation reaction at interfaces. The desorption temperature of CO on Pt$_n$/CeCu is much lower than that on Pt$_n$/Ce, indicating the weakening of interfacial CO adsorption by Cu dopants. Thus, the interfacial oxygen site of Pt$_n$/CeCu is activated and CO adsorption strength at the interface is weakened, contributing to the unique interfacial structure and reducibility in the Pt sub-nanoclusters and Cu-doped CeO$_2$ support composites.

To further clarify the interfacial structures of supported Pt catalysts, X-ray absorption fine structure (XAFS) spectra tests were performed. The white line intensity of Pt L$_{III}$-edge XANES spectra in Fig. 3a indicates that oxidized Pt species exist in all composite catalysts, qualitatively indicating electron transfer between Pt and oxide supports. The white line intensity of the catalysts follows the order: Pt$_1$/CeCu > Pt$_1$/Ce > Pt$_n$/Ce ≈ Pt$_n$/CeCu, implying that Pt single atoms transfer larger numbers of electrons to oxide supports than Pt sub-nanoclusters. Moreover, the white line intensities of Pt$_1$/CeCu and Pt$_1$/Ce are close to that of PtO$_2$, indicating that the supported Pt single atoms are mainly in oxidized states. The white line intensity of Pt$_n$/Ce is similar to that of Pt$_n$/CeCu, implying that the valence states of Pt catalysts are dependent on the morphologies, which is consistent with the Pt 4$f$ XPS analysis (Supplementary Fig. 23 and Supplementary Table 2). The Fourier transform extended X-ray absorption fine structure (EXAFS) spectra of Pt$_1$/CeCu and Pt$_1$/Ce in Fig. 3b show one prominent peak at about 1.7 Å, which is assigned to the

formed Pt–O bond between Pt single atoms and oxide supports. The EXAFS spectra of Pt$_n$/CeCu and Pt$_n$/Ce exhibit a decrease in signal from the Pt–O contribution and very weak signal at the range of Pt–Pt bonds compared to Pt foil. The small Pt–Pt contribution can be attributed to the low crystallinity and high disorder of Pt atoms for the deposited Pt clusters at relatively low temperature as presented in the HADDF-STEM images, which agrees with previously reports[20,40].

By combining the results of HAADF-STEM images and EXAFS spectra, stable interface structures of Pt single atoms and Pt$_5$ clusters supported on CeO$_2$ (110) slabs (Pt$_1$/Ce and Pt$_5$/Ce) were constructed and optimized by density functional theory (DFT) calculations (Supplementary Figs. 24, 25). Pt$_1$/CeCu and Pt$_5$/CeCu models have an interfacial Cu dopant replacing a Ce atom (Supplementary Fig. 25). As shown in Fig. 3c, the calculated Bader charges[47] of interfacial Pt atoms (Q$_{Pt}$) show the same trend as the white line intensity that are related to electron transfer at the interface. Moreover, bond lengths between Pt atoms and interfacial oxygen (L$_{Pt-O}$), and coordinate number of interfacial Pt atoms (CN$_{Pt-O}$) also show the same trend as L$_{Pt-O}$, and CN$_{Pt-O}$ from the fitting data of EXAFS results (Supplementary Fig. 26 and Supplementary Table 3), validating the constructed models. The formation of Pt sub-nanoclusters on Pt$_n$/CeCu and Pt$_n$/Ce leads to lengthened Pt–O bonds and decreased CN$_{Pt-O}$, which is associated with decreased electron transfer compared to Pt$_1$/CeCu and Pt$_1$/Ce. The relatively small CN$_{Pt-O}$ values of Pt$_n$/Ce and Pt$_n$/CeCu can be related to the highly catalytic activity compared with Pt$_1$/Ce and Pt$_1$/CeCu, respectively[29].

**CO oxidation mechanism studies.** The adsorption energies of CO on the supported Pt single atom and cluster have been

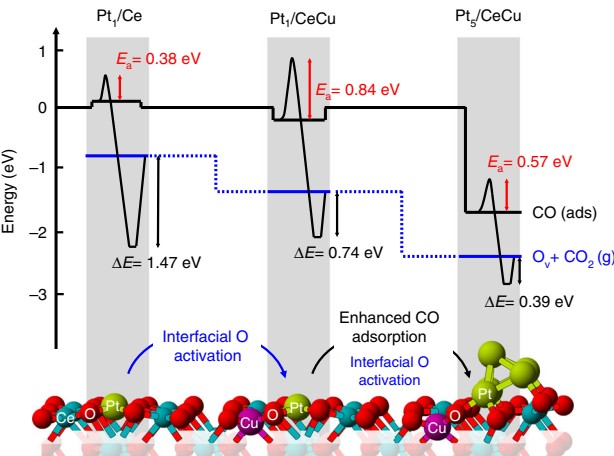

**Fig. 4 Oxidation energetics of adsorbed CO molecules by interfacial oxygens of Pt$_1$/Ce, Pt$_1$/CeCu and Pt$_5$/CeCu.** Source data are provided as a Source Data file.

calculated as shown in Supplementary Fig. 27. The top Pt atoms for both Pt$_5$/Ce and Pt$_5$/CuCe will be poisoned by CO molecule with overbinding that is consistent with our DRIFTS results, i.e. some CO molecules remained after O$_2$ flow was introduced (Fig. 2d). Therefore, we investigated the CO oxidation processes at the interface of catalysts based on the Mars-van Krevelen mechanism[48], where adsorbed CO molecules first react with interfacial lattice oxygen along creating an oxygen vacancy (O$_v$). The formation energies of O$_v$ at the interface of our constructed models have been presented in Supplementary Fig. 28. According to the CO adsorption energies and calculated oxygen vacancy formation energies, the energies of the initial states with an adsorbed CO molecule (black line) and the final states with oxygen vacancy after CO$_2$ molecule desorption (blue line) are presented in Fig. 4. Although CO adsorbed on Pt$_1$/Ce can easily react with the lattice oxygen by overcoming a low energy barrier (0.38 eV), the CO$_2$ product at the interface (Supplementary Fig. 29) is very stable with a desorption energy of 1.47 eV, thus poisoning the active sites for subsequent reactions. The formation of interfacial Cu–O–Ce site in doped sample can decrease the desorption energy of formed CO$_2$ to 0.74 eV for Pt$_1$/CeCu, which is associated with the activated oxygen in Cu–O–Ce site. However, it needs to overcome an energy barrier of 0.84 eV to form the CO$_2$ intermediate at the interface due to the weak adsorption of a CO molecule on Pt single atoms (−0.26 eV).

After Pt single atoms are aggregated to a Pt$_5$ cluster, the energy barrier is decreased to 0.57 eV due to the appropriately enhanced CO adsorption strength (−1.73 eV). Moreover, the more activated oxygen in the interfacial Cu–O–Ce site leads to a decrease in CO$_2$ desorption energy (0.39 eV). However, in terms of Pt$_5$/Ce (Supplementary Fig. 30), a high energy barrier of 1.10 eV is needed to overcome the strong CO poisoning effect on Pt clusters (−2.17 eV). The high energy barrier for Pt$_5$/Ce also agrees well with the large activation barrier of Pt$_n$/Ce in the kinetic test, which results from the strong CO poisoning effect on the subnanometric Pt clusters. The differential charge densities indicate that the bonding between interfacial Pt and oxygen is strengthened with an enhanced polarization due to the low-valence Cu dopant, which leads a weakened adsorption of CO molecule at interfacial Pt atoms (Supplementary Fig. 31). In order to eliminate potential variations due to the size of Pt clusters used in the calculation, the energetic routes of CO oxidation at the interfaces of oxide slabs supported a larger Pt$_{14}$ cluster have also been calculated (Supplementary Fig. 32), which exhibit the similar results to supported Pt$_5$ clusters. In addition, the oxidation

processes of co-adsorbed CO and O$_2$ molecules at interfaces containing an O$_v$ show that Pt$_5$/CeCu exhibits the highest activity with the lowest energy barrier of 0.53 eV (Supplementary Fig. 33). Therefore, both activated interfacial oxygen and moderate CO adsorption strength are keys to the excellent catalytic activity of Pt$_5$/CeCu.

## Discussion

Precisely controlled interfacial structures of Pt/CeO$_2$ have been proposed by regulating oxide doping and Pt cluster size and employing the redox-coupled ALD method. Cu dopants have been introduced to modulate the surface reducibility of CeO$_2$ support, which led to the formation of Cu$^{2+}$ ions in the CeO$_2$ lattice and helped create active Cu–O–Ce sites for tuning the interfacial structures and anchoring the Pt catalysts. The concentration of Cu dopants is kept low to avoid the segregation of CuO$_x$ species that are unfavorable for the activity enhancement of supported Pt catalysts. The size of deposited Pt catalysts in the range of sub-nanometer clusters to single atoms are well controlled by our proposed ALD recipe, which can directly affect the coordination environment of interfacial Pt atoms. The supported Pt single atoms with larger coordination number of Pt–O show lower activity than subnanometric Pt clusters with smaller coordination number of Pt–O[29,30]. Compared with subnanometric Pt clusters on CeO$_2$ supports, Cu dopants at the interfaces can not only activate the interfacial oxygen, but also weaken the adsorption of CO molecule at interfacial Pt atoms, which are the keys to promoting CO oxidation with lattice oxygen at room temperature in Pt/CeO$_2$ catalysts[45,46]. The experimental evidence agrees well with our proposed catalytic reaction path at the interfacial sites following the Mars-van Krevelen mechanism based on DFT calculations. The resulting Pt sub-nanoclusters on Cu-doped CeO$_2$ exhibit excellent activity with an onset of CO oxidation reactivity below room temperature. Our study lays the foundation for designing and preparing highly efficient Pt catalysts for low-temperature exhaust abatement.

## Methods

**Catalysts preparation.** The CeO$_2$ nanorod supports were prepared by hydrothermal method. In details, we first dissolved Ce(NO$_3$)$_3$·6H$_2$O and NaOH (1.96 and 16.88 g, Sinopharm Chemical Reagent Co., Ltd) into 40 ml and 30 ml of deionized water, respectively. After both solutions were cooled down to room temperature, the NaOH solution were slowly dropped into the Ce(NO$_3$)$_3$·6H$_2$O solution with continuously magnetic stirring. Then, the mixture solution was poured into a 100 ml Teflon bottle, which was subsequently sealed in a stainless steel vessel autoclave. The hydrothermal reaction was performed at 100 °C for 24 h. After centrifugation collection, repetitive wash and vacuum dry (80 °C), the CeO$_2$ precursors were calcined at 500 °C for 4 h to obtain the final CeO$_2$ nanorod supports. In order to tune the reducibility of CeO$_2$ supports, Cu dopants were introduced by substituting part of Ce(NO$_3$)$_3$·6H$_2$O with a certain amount of Cu(NO$_3$)$_2$·3H$_2$O (Sinopharm Chemical Reagent Co., Ltd) before being dissolved into deionized water. The molar ratio of Ce and Cu was kept as 99:1. The finally obtained Cu-doped CeO$_2$ nanorod support was denoted as Ce$_{0.99}$Cu$_{0.01}$O$_2$. As a comparison, the Cu-doped CeO$_2$ nanorod support with the Ce/Cu molar ratio of 95:5 was also prepared, which was denoted as Ce$_{0.95}$Cu$_{0.05}$O$_2$.

The Pt single atoms and sub-nanoclusters were loaded on CeO$_2$ and Ce$_{0.99}$Cu$_{0.01}$O$_2$ nanorod supports via ALD method. The ALD processes were performed in a custom-made fluidized-bed reactor (AngstromBlock Scale-F015 ALD system) as described in our previous studies[34,35]. For each ALD process, about 200 mg nanorod supports were loaded in a designed holder, which were firstly fluidized by 200 mL min$^{−1}$ of N$_2$ at 150 °C for 30 min. The Pt single atoms were loaded on CeO$_2$ and Ce$_{0.99}$Cu$_{0.01}$O$_2$ nanorod supports after one cycle of Pt ALD with trimethyl(methylcyclopentadienyl)platinum (MeCpPtMe$_3$, 98%, Sigma-Aldrich) and ultrahigh purity O$_2$ as precursors. The MeCpPtMe$_3$ precursor, which was kept at 65 °C during Pt ALD, was introduced into the reactor at 150 °C with the pulse time and purge time of 100 s and 100 s. Then, the reactor was rapidly heated to 200 °C. 500 mL min$^{−1}$ of O$_2$ was subsequently introduced into the reactor with the pulse time and pure time of 600 and 200 s. For the preparation of Pt$_n$/Ce and Pt$_n$/CeCu, one cycle of redox-coupled Pt ALD was performed. The processes of Pt precursor and O$_2$ were the same as that of Pt single atom preparation. After the purge step of O$_2$, 500 mL min$^{−1}$ of ultrahigh purity H$_2$ was

subsequently introduced into the reactor at 200 °C with the pulse time and purge time of 600 and 200 s.

**Characterizations.** The morphology of $CeO_2$ nanorod supports was characterized by transmission electron microscopy (TEM, Tecnai G2 F30 electron microscope, FEI). The aberration-corrected high-angle annular dark-field scanning transmission electron microscopy (HAADF-STEM) images and energy dispersive spectroscopy (EDS) spectra were obtained on a JEOL JEM-ARM200F TEM. The X-ray diffraction (XRD) patterns were recorded by the PANalytical X'Pert Pro with a Cu Kα1 radiation source. The mass loading of Pt on catalysts was analyzed by inductively coupled plasma atomic emission spectrometer (ICP-OES) on Optima 4300 DV spectrometer. The Raman spectra were recorded on a Renishaw inVia Reflex in a range from 200 to 800 $cm^{-1}$ using an excitation laser line of 532 nm. The X-ray photoelectron spectroscopy (XPS) spectra were obtained with an AXIS-ULTRA DLD-600W XPS spectrometer in high vacuum environment (~$10^{-7}$ Pa) after all samples are kept in the XPS chamber overnight, which were calibrated by the peak of C 1$s$ at 284.8 eV.

The in situ diffuse reflectance infrared Fourier transform spectroscopy (DRIFTS) spectra of CO on prepared catalysts were collected by wide band mercury cadmium telluride detector on the Nicolet iS50 FTIR spectrometer (ThermoFisher Scientific). The Praying MantisTM diffuse reflection accessory (Harrick Scientific Products Inc.) was used with ZnSe windows. The catalysts were first pretreated by 30 mL $min^{-1}$ of ultrahigh purity $N_2$ at room temperature for 30 min, and the background spectra were collected. Then, the gas was switched to 30 mL $min^{-1}$ of 1% vol. CO balanced by $N_2$. The CO DRIFTS spectra were collected after 10 min. In order to study the adsorption strength of CO on catalysts, the gas was subsequently switched to 30 mL $min^{-1}$ of ultrahigh purity He. The DRIFTS spectra of CO adsorption on catalysts were collected every 2 min.

The temperature-programmed reduction by hydrogen ($H_2$-TPR) was performed by a chemisorption analyzer (AMI-300 series, Altamira Instrument). Typically, 30 mg of the catalyst was supported by quartz wool in a U-type quartz tube reactor, which was pretreated using 30 mL $min^{-1}$ of Ar at 100 °C for 30 min. The feed was switched to 30 mL $min^{-1}$ of 10% vol. $H_2$ balanced with Ar, when the catalyst was cooled down to room temperature. Then, the reactor was heated to 800 °C with a ramp rate of 5 °C $min^{-1}$ and thermal conductivity detector was utilized to monitor the signal of $H_2$ consumption.

The temperature-programed desorption of CO (CO-TPD) was performed by the VDSorb-91x chemisorption analyzer. 100 mg of the catalyst was supported by quartz wool in a U-type quartz tube reactor. After being pretreated using 50 mL $min^{-1}$ of He at 200 °C for 30 min, the catalyst was cooled down to −100 °C by liquid nitrogen trap under He flow. The catalyst was exposed to 50 mL $min^{-1}$ of 10 vol. % CO balanced by He for 30 min, then the feed was switched to He to purge the catalyst until the baseline was stable. CO-TPD curves were obtained under the He flow by using the AMETEK® quadrupole mass spectrometer to monitor the signal of CO ($m/z = 28$) and $CO_2$ ($m/z = 44$), when the reactor was heated to 600 °C with a ramp rate of 5 °C $min^{-1}$.

The X-ray absorption fine structure (XAFS) spectra were obtained at the 1W1B beamline of Beijing synchrotron radiation facility. The incident photon beam was selected by a double-crystal Si (111) monochromator after a collimating mirror and focused by a toroidal mirror. All XAS measurements were conducted in transmission mode using a 19-element high-purity germanium solid-state detector. The X-ray beam size on our prepared catalysts was about $0.9 \times 0.3$ $mm^2$ at half-maximum (FWHM) with a photon flux of >$4 \times 10^{11}$ photons $s^{-1}$ at 9 keV. Pt $L_{III}$-edge in the energy range of 11368–12463 eV was collected for catalysts. In order to avoid the influence of air, all catalysts were tested under $N_2$ flow at room temperature in the cell as our previous study reported[34]. The Pt $L_{III}$-edge of Pt foil and $PtO_2$ were also tested as references. By considering the parameters of coordinated number (N), bond length (R, Å), Debye–Waller factor ($σ^2$, $Å^2$) and shift in the edge energy ($ΔE_0$, eV), the Pt $k^2$-weighted Fourier-transformed extended X-ray absorption fine structure (EXAFS) spectrum of catalysts was fitted by using the Pt foil and $PtO_2$ modes in Demeter program[49].

**Activity evaluation.** The CO oxidation activity evaluation of catalysts was performed by the VDSorb-91x chemisorption analyzer. 50 mg of catalysts without any pre-treatments were packed between quartz wool in a U-type quartz tube reactor. The flow rate of feed gas was 100 mL $min^{-1}$, which was a mixture of CO (1% vol.), $O_2$ (10% vol.) and $N_2$. The temperature of reactor was linearly heated from −20 to 200 °C with a ramp rate of 2 °C $min^{-1}$. The analysis of outlet gas mixture was performed by in situ HPR-20 mass spectrometer, which tested the partial pressure of $CO_2$ ($p_{CO_2}$). The CO conversion ($X_{CO}$) was calculated by

$$X_{CO} = \frac{p_{CO_2}^T - p_{CO_2}^{start}}{p_{CO_2}^{end} - p_{CO_2}^{start}} \times 100\%, \tag{1}$$

where $p_{CO_2}^{start}$, $p_{CO_2}^{end}$, and $p_{CO_2}^T$ are the partial pressure of $CO_2$ in the outlet gas mixture before CO oxidation, after the total conversion of CO and at a reaction temperature (T), respectively. $p_{CO_2}^{start}$ and $p_{CO_2}^{end}$ were calibrated by the portable emission analyzer (MEXA-584L, Horiba). The kinetics tests were performed with CO conversion below 15% to eliminate the thermal and diffusion effects. The inert quartz was used to dilute the catalysts to make sure the appropriate CO conversion. The turnover

frequency (TOF) of catalysts at a reaction temperature ($T$) was tested by decreasing the mass of catalysts, which was calculated by

$$TOF = \frac{\frac{P \times V}{R \times T} \times w_{CO} \times X_{CO}}{n_{Pt}}, \tag{2}$$

where $P$, $V$, $R$ and $w_{CO}$ were the pressure of feed gas (101325 Pa), the flow rate of feed gas ($1.67 \times 10^{-6}$ $m^3$ $s^{-1}$), universal gas constant (8.134 J $mol^{-1}$ $s^{-1}$) and vol. percentage of CO (1%), respectively. $n_{Pt}$ was the mole mass of Pt in catalysts determined by ICP-OES method. According to the slope of the Arrhenius plots, the activation energies ($E_a$) of catalysts were calculated.

**DFT calculations.** Spin-polarized calculations were carried out based on DFT with the Vienna Ab initio Simulation Package (VASP) using the Perdew–Burke–Ernzerhof (PBE) functional[50–53]. The ionic cores were described by projector augmented wave method[54]. In order to correct the on-site Coulomb and exchange interactions, PBE+U method with U-J = 5.0 eV was applied to describe Ce 4$f$-orbital[55,56]. The energy cutoff of plane wave basis was set to 400 eV. The convergence criterion of atomic structural optimization was 0.05 eV $Å^{-1}$. The $CeO_2$ (110) slab was built with vacuum layer thicknesses larger than 15 Å to avoid the interaction between two periodic slabs. The Brillouin zone was sampled at the Γ point. The climbing image nudged elastic band (CI-NEB)[57] method with six intermediate images was used to calculate the minimum energy paths of CO oxidation at the interfaces of $Pt/CeO_2$ catalysts.

## Data availability
The source data underlying Figs. 2-4 and Supplementary Figs. 2–4, 7–23, 25, 26, 31 are provided as a Source Data file. Extra data are available from the corresponding authors upon request.

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

## Acknowledgements

This work was supported by the National Natural Science Foundation of China (Grants No. 51871103, 51835005, 51801067, and 51911540476) and Program for HUST Academic Frontier Youth Team (2019kfyXMBZ025 and 2018QYTD03). Xiao Liu gratefully acknowledges the support from the Postdoctoral Innovation Talents Support Program (BX20180104). Ming Yang acknowledges the support of General Motors Research and Development and the start-up funding provided by Clemson University. We would also like to acknowledge the technology support from the Analytic Testing Center, Flexible Electronics Research Center and SCTS/CGCL HPCC of HUST.

## Author contributions

X.L. conducted the synthesis and characterizations of catalysts, and wrote the paper. S.F.J. and J.B.W. performed HAADF-STEM characterizations. X.L. and Y.T.T. performed catalytic activity evaluation. X.L. and Y.W.W. were responsible for the detailed DFT calculations. S.Q.C. performed XAFS characterizations. M.Y., B.S. and R.C. directed the project and mentored the paper writing. All authors discussed the results and commented on the paper.

## Competing interests

The authors declare no competing interests.
