## [Peer Review File · Nature Communications]

Reviewers' Comments:

Reviewer #1:

Remarks to the Author:

The manuscript of Liu et al. describes the promotional effect of copper on the CO oxidation activity of platinum-ceria catalysts with controlled dispersion in sub-nm range prepared by novel state-of-the-art method. Spectroscopic characterization performed unfortunately only ex situ suggest that the redox activity of cerium can be involved in the mechanism. The adsorption of CO also gets weaker in the presence of copper according to in situ DRIFTS. Theoretical calculations confirm that both effects can be beneficial for the catalytic activity.

The manuscript is of potential interest for the wide community of scientists in the field of catalysis and design of nanomaterials. At the same time, there are several critical points, which do not allow saying that all conclusions of this work are supported by the experimental or theoretical data. Therefore, major revision is recommended. Addressing the following points can significantly improve understanding of the effects reported of this work and clarify their significance:

1. Figure 2a shows the effect of Cu on the CO conversion of Pt1/Ce and Ptn/Ce catalysts. The strong positive effect of Cu on the CO conversion temperature is misleading as samples containing Cu also contain 2.5 higher concentration of Pt. Figure S8 illustrates that by diluting the catalysts the effect of Cu gets smaller but is still there. At the same time, it is not explained why the catalyst for these tests were diluted only by a factor of 1.6 and not by 2.5 based on Pt loading and similar Pt dispersion within the error bar. This observation questions whether the effect of Cu exists and whether it is significant considering the error bars on concentration of Pt and dispersion of Pt.
2. Kinetic characterization of the catalyst activity at low conversion regime requires additional details describing experiments and may be additional tests showing reproducibility. What kind of pre-treatment was used before measuring the activity of the catalysts to remove adsorbed species such as carbonates? Which material was used to dilute the catalyst during tests? Concerning Figure 2b, was the activity measured after initial deactivation tests of the catalysts? Were the activity data and the activation energies reproducible upon temperature cycling? The activation energy for Ptn/Ce seem to be very high, what can be the reason? For comparison, it would be useful to report also the activity of CuCe catalyst in the Figure 2b.
3. The formula in the line 363 contains parameter w corresponding to the fraction of Pt atoms at Pt-support interface, which are also called perimeter sites. This parameter is used for TOF calculation but there are no details how this parameter was estimated for each catalyst. Besides, Table S1 compares TOFs for different Pt containing catalysts measured in this work with those previously reported in the literature. Can the authors confirm that in all these works TOF parameter was estimated based on the number of above-mentioned perimeter sites and that this number was estimated in a similar way?
4. The authors report (based of the DRIFT data) that CO absorption strength gets lower for

Ptn/CuCe in comparison to Ptn/Ce catalyst. This effect is interesting and novel. It would make sense to confirm it by TPD experiments. Besides, measuring of the reaction orders for CO and oxygen would be useful, as weaker CO adsorption on Ptn/CuCe can affect the reaction mechanism in different ways. May be reaction on Ptn/CuCe takes place not only at the interface but also on Pt nanoparticles surface due to weaker CO adsorption?

5. The EXAFS analysis involving second coordination shell of Pt reported in Table S2 does not look very reliable: the fitted curves are not presented, the fitted ΔE of 21 -29 eV is too high, which suggests that fits might not be correct. Besides, basic description of XAS beamline is missing: type of monochromator, mirrors, detectors, flux, beam size...

6. Figure 3c compares the calculated Bader charges of interfacial Pt atoms to the structural parameters of Pt determined by XAS measured ex situ. What is the origin of these correlations from the theoretical point of view? Can the authors add the error bars to the plots to confirm that the changes are significant? How relevant are the reported correlations considering that in air most of Pt atoms in Ptn particles are oxidized by air, while during catalysis they should be partially reduced, thus large fraction of oxygen in their local coordination should be replaced by CO?

7. Concerning XPS measurements, were they performed in vacuum? Was the possibility of photo-reduction of Ce⁴⁺ during the measurements considered?

8. The term "modulation" used in the manuscript title seems to be not exact for the reported phenomena.

Reviewer #2:

Remarks to the Author:

The authors have studied CO oxidation on Pt deposited on Cu doped ceria nanorods. The results show high reactivity for Pt sub-nano clusters for CO oxidation. There is considerable interest in improving the reactivity of Pt catalysts for this reaction, so the work is potentially interesting. However, the authors have overlooked some recent literature, which causes me to question their interpretation. The approach they used to report reactivity, TOF based on interfacial sites, is not consistent with standard practice in this field. Finally, they state that Cu doped ceria is not active for CO oxidation at low temperatures, which is not true. For these reasons, I do not think the manuscript is not suitable for publication in its present form.

1) Cu doped ceria is known to be active for CO oxidation as seen in reference 1 and 2 below. Atomically dispersed Cu shows onset of CO oxidation reactivity starting at room temperature.

1. W.-Z. Yu, W.-W. Wang, S.-Q. Li, X.-P. Fu, X. Wang, K. Wu, R. Si, C. Ma, C.-J. Jia, and C.-H. Yan, Construction of Active Site in a Sintered Copper–Cerium Nanorod Catalyst *Journal of the American Chemical Society* 141 (44) (2019) 17548-17557 DOI: 10.1021/jacs.9b05419.

2. K. Kappis, C. Papadopoulos, J. Papavasiliou, J. Vakros, Y. Georgiou, Y. Deligiannakis, and G. Avgouropoulos, Tuning the Catalytic Properties of Copper-Promoted Nanoceria via a Hydrothermal Method *Catalysts* 9 (2) (2019) 138 DOI:

2) The authors state that the adsorbed CO is lost when flowing He after CO oxidation (Figure 2d). A similar observation was reported in reference 3 where the sample does not contain any Cu. Furthermore, reference 4 shows clearly that this loss of adsorbed CO is a result of oxygen being supplied to the Pt clusters from the ceria support. It is not a result of weakening of the metal-CO bond. Hence, the authors need to consider these references and revise their explanation.

3. X.I. Pccira-Hcrnandcz, A. DeLaRiva, V. Muravev, D. Kunwar, H. Xiong, B. Sudduth, M. Engelhard, L. Kovarlk, E.J.M. Hcnscn, Y. Wang, and A.K. Datye, Tuning Pt-CeO₂ interactions by high-temperature vapor-phase synthesis for improved reducibility of lattice oxygen *Nature Communications* 10 (2019) DOI: 10.1038/s41467-019-09308-5.

4. Y. Lu, C. Thompson, D. Kunwar, A.K. Datye, and A.M. Karim, Origin of the High CO Oxidation Activity on CeO₂ Supported Pt Nanoparticles: Weaker Binding of CO or Facile Oxygen Transfer from the Support? *Chemcatchem* (2020) DOI: 10.1002/cctc.201901848.

3) They report a turnover frequency calculated using a term w interface (line 363). This is the ratio of active Pt atoms to total Pt atoms. It is not clear how this number is calculated. It is notoriously difficult to pinpoint the number of interface sites, if those are the only ones active. Even reference 16 by Cargnello et al. who focused on the role of interfacial sites based their reactivity on total Pt atoms, and only used a model to make their case that corner atoms are likely to be active. The standard practice in the literature on single atom catalysts is to simply report the TOF based on total Pt atoms. Hence the authors need to use the more conventional definition and then compare their reactivity with state of the art Pt-ceria catalysts, such as those in reference 3 above or reference 5 below.

5. H. Wang, J.-X. Liu, L.F. Allard, S. Lee, J. Liu, H. Li, J. Wang, J. Wang, S.H. Oh, W. Li, M. Flytzani-Stephanopoulos, M. Shen, B.R. Goldsmith, and M. Yang, Surpassing the single-atom catalytic activity limit through paired Pt-O-Pt ensemble built from isolated Pt-1 atoms *Nature Communications* 10 (2019) DOI: 10.1038/s41467-019-11856-9.

Reviewer #3:

Remarks to the Author:

This paper proposes a redox-coupled ALD method to regulate oxide doping and Pt cluster size of Pt/CeCu catalysts. The synthesis might be interesting. However, subnanometric Pt anchored on CeO₂ with the best catalytic activity for CO oxidation has been widely reported in previous reports, such as *ACS Catalysis* 2015, 5, 5164-5173. Also, Cu-doped CeO₂ itself is also a good catalyst for low temperature of CO oxidation. In this case, the novelty of this

work is not competitive for Nature Communications. In the catalytic mechanism, the in-situ experiments are expected to monitor the evolution of cerium, copper, platinum as well as oxygen vacancy. Considering the very small size of Pt cluster and the strong metal-support interaction between Pt cluster and Cu-doped CeO₂, the dynamic changes of their chemical environments are important to understand the catalytic mechanism. The current evidences are not convincing enough. I cannot suggest the publish of this work. For other comments:

1. Chemical status of Cu, in the lattice of CeO₂ or as CuO_x clusters? Also provide the evidences. Then, how about the spatial distribution of Pt cluster on CeO₂ and Cu-CeO₂?
2. In the synthesis, can the Pt mass loading of Pt/CeCu be controlled?
3. The reducibility of all catalysts should be characterized by temperature-programmed reduction.
4. In situ DRIFTS spectra, I doubt on the desorption of CO on the catalysts by He sweeping. Generally, the chemical adsorbed CO on Pt quite strong. Sweeping at room temperature might not be enough. I think the CO-temperature programmed desorption should be correct choice to evaluate the adsorption strength of CO on various catalysts.
5. In their DFT calculation, Pt₅ metallic cluster was used. In the TEM images, the number of Pt atoms in cluster is over 30. To build a model over 30 might be not practical for DFT calculation. However, Pt₅ is quite small cannot match well with experiments. Especially, only two layers of Pt with four Pt atom with support and one Pt atom on the top, it cannot reflect the true oxidation state of PtO_x clusters. Also, the Pt in their real catalysts was identified as the oxidized Pt (X-ray absorption data). Also, I checked their experimental section, where no pre-reduction by H₂ had been mentioned. In my opinions, three layers of PtO_x cluster with more Pt should be built on the top of supports for DFT calculations. In this case, I cannot trust their DFT calculations, in which the model did not match the experiments.
6. In the Supplementary Figure 6, why is Cu so much less than Pt?
7. Comparing the results of Figure 3 and Supplementary Figure 14, why was no Pt⁴⁺ find from the Pt 4f XPS analysis?

Response to reviewers

Reviewers' comments:

Reviewer #1:

The manuscript of Liu et al. describes the promotional effect of copper on the CO oxidation activity of platinum-ceria catalysts with controlled dispersion in sub-nm range prepared by novel state-of-the-art method. Spectroscopic characterization performed unfortunately only ex situ suggest that the redox activity of cerium can be involved in the mechanism. The adsorption of CO also gets weaker in the presence of copper according to in situ DRIFTS. Theoretical calculations confirm that both effects can be beneficial for the catalytic activity.

Author reply:

We thank the reviewer for taking his/her time evaluating our manuscript and agree with the reviewer that the catalytic mechanism studies of our prepared catalysts can be strengthened by adding certain *in situ* characterizations. Additional *in situ* DRIFTS experiment of CO oxidation of Pt_n/CeCu catalyst has been performed at room temperature to investigate the catalytic mechanism. As shown in Fig. R1, the signals of adsorbed CO molecules in the *in situ* DRIFTS spectra of Pt_n/CeCu decrease when the flow is switched from 30 mL/min of He to 30 mL/min of 1% vol. O₂ balanced by N₂. Meanwhile, the appearance of stretching bands of CO₂ molecule at 2360 cm⁻¹ and 2330 cm⁻¹ suggests that the adsorbed CO molecules on Pt_n/CeCu can be further oxidized by activated O₂ molecules at the interface, which agrees well with our theoretical results (Supplementary Fig. 31) and previous literatures on Pt/CeO₂ catalysts. (*Nat. Commun.* 2019, 10: 1358; *ChemCatChem* 2020, 12, 1726). In the revised manuscript, we have incorporated these new results and corresponding references to strengthen the catalytic mechanism discussion of our prepared catalysts.

Figure R1. *In situ* DRIFTS spectra of CO adsorption and oxidation of Pt_n/CeCu. After CO exposure, He flow is continued and the spectra are recorded at 10 min. Subsequently, the flow is switched to 1% vol. O₂ balanced by N₂ and the spectrum is recorded at 2 min.

Modification:

1. Added the discussion on *in-situ* DRIFTS of CO oxidation of Pt_n/CeCu. “When the flow is switched from He to O₂, the signal of adsorbed CO molecules is further weakened. The appearance of stretching bands of CO₂ molecule at 2360 cm⁻¹ and 2330 cm⁻¹ suggests that the adsorbed CO molecules on Pt_n/CeCu can be further oxidized by activated O₂ molecules at the interface, which agrees well with the previous studies on Pt/CeO₂ catalyst^{45,46}.”

(First paragraph in Page 10 in text)

2. Revised Fig. 2d and added Supplementary Fig. 19.

(Fig. 2d and Supplementary Fig. 19)

The manuscript is of potential interest for the wide community of scientists in the field of catalysis and design of nanomaterials. At the same time, there are several critical

points, which do not allow saying that all conclusions of this work are supported by the experimental or theoretical data. Therefore, major revision is recommended. Addressing the following points can significantly improve understanding of the effects reported of this work and clarify their significance:

Author reply:

We appreciate the reviewer's recognition on the importance of this topic to the wide catalysis community. In the revised manuscript, we made our best efforts to improve the work by adding new experimental results, such as *in situ* characterizations, repeatability tests, reaction order tests, CO-TPD and H₂-TPR. The detailed experimental data and discussion have been presented below in response to listed comments. We believe that we have addressed the reviewer's concerns and improved the quality of our work. We hope the revised manuscript will deem fit to be published in *Nature Communications*.

1>> Figure 2a shows the effect of Cu on the CO conversion of Pt1/Ce and Ptn/Ce catalysts. The strong positive effect of Cu on the CO conversion temperature is misleading as samples containing Cu also contain 2.5 higher concentration of Pt. Figure S8 illustrates that by diluting the catalysts the effect of Cu gets smaller but is still there. At the same time, it is not explained why the catalyst for these tests were diluted only by a factor of 1.6 and not by 2.5 based on Pt loading and similar Pt dispersion within the error bar. This observation questions whether the effect of Cu exists and whether it is significant considering the error bars on concentration of Pt and dispersion of Pt.

Author reply:

We thank the reviewer for carefully checking detailed experimental records. Indeed, we found that the weights of supported Pt catalysts and diluted Ce_{0.99}Cu_{0.01}O₂ supports have been labeled incorrectly in the caption of Supplementary Fig. 11. In order to eliminate the effects of Pt's mass loading, we diluted 20 mg Pt_n/CeCu or Pt₁/CeCu with 30 mg Ce_{0.99}Cu_{0.01}O₂, which corresponded to the factor of 2.5 for the decrease of Pt's mass loading. We thank the reviewer for pointing out our typo and

have revised the corresponding description of Supplementary Fig. 11. In terms of the concentration and dispersion of Pt, we agree with the reviewer that they are important for accurate activity evaluations of our prepared catalysts. To this end, the ICP-OES characterization has been performed to test Pt's mass loading variation of three samples of Pt_n/CeCu that were prepared by the same process as described in our manuscript. The average mass loading of Pt for Pt_n/CeCu is 1.77 ± 0.11 wt% based on the ICP-OES results (1.65 wt%, 1.79 wt% and 1.86 wt%). The consistency of Pt's mass loading can be attributed to the well-controlled process of atomic layer deposition method. The Pt size distribution analysis of AC-STEM images (Supplementary Fig. 5) indicate that the sizes of Pt clusters in Pt_n/CeCu and Pt_n/Ce are similar. Therefore, we believe the slight variation on concentration and dispersion of Pt towards the activity evaluation of our prepared catalysts are negligible compared with the effect of Cu dopants.

Modification:

1. Revised the caption of Supplementary Fig. 11. *“20 mg Pt_n/CeCu or Pt₁/CeCu has been diluted with 30 mg Ce_{0.99}Cu_{0.01}O₂ supports to make sure the mass loading of Pt similar to Pt_n/Ce or Pt₁/Ce catalysts.”*

(Supplementary Fig. 11 in SI)

2. Revised the mass loading of Pt_n/CeCu in Supplementary Table 1 and the calculated TOF value.

(Supplementary Table 1 in SI)

2>> Kinetic characterization of the catalyst activity at low conversion regime requires additional details describing experiments and may be additional tests showing reproducibility. What kind of pre-treatment was used before measuring the activity of the catalysts to remove adsorbed species such as carbonates? Which material was used to dilute the catalyst during tests? Concerning Figure 2b, was the activity measured after initial deactivation tests of the catalysts? Were the activity

data and the activation energies reproducible upon temperature cycling? The activation energy for Pt_n/Ce seem to be very high, what can be the reason? For comparison, it would be useful to report also the activity of CuCe catalyst in the Figure 2b.

Author reply:

We appreciate the reviewer's comments on the kinetic tests of supported Pt catalysts and have added relevant description of our experiments in details, as well as performed additional tests to address the reviewer's concern. In terms of the question on the pre-treatment of catalysts, the catalyst is tested as is after the ALD process, without deliberately performing additional pre-treatments. The catalyst is quite stable with respect to cycling tests with negligible change in light off temperature (shown in Fig. R2), implying the robustness of the fabrication process and reproducibility of the catalytic performance results. The negligible shifts of CO conversion curves also indicate the structural stability of our constructed Pt/oxide interface, which can be attributed to the gas-phase based atomic layer deposition method that has minimal detrimental effect on the surface structure of oxide supports as reported in our previous work (*Chem. Sci.* 2018, 9, 2469).

Figure R2. CO conversion curves of Pt_n/CeCu catalyst for three cycles. The slightly shifts of CO conversion curves indicate the reproducibility of our catalytic results.

In the kinetic tests, we diluted the catalysts with inert quartz sand consistent with

previous literatures (*J. Am. Chem. Soc.* 2011, 133, 4498; *Science* 2013, 341, 771). We described the catalytic evaluation method in more details in the revised manuscript. In our work, the activation energy of Pt_n/Ce is determined to be 97.62 kJ/mol, which can be attributed to the strong CO poisoning effect on the subnanometric Pt clusters. The experimental result agrees with the large barrier energy of CO oxidation at the interface of Pt_n/Ce in our DFT calculations. Since the activation energy is related to a number of factors, including size, morphology and chemical states of supported Pt catalysts, we believe our relatively high activation energy of ALD prepared Pt_n/Ce catalyst is reasonable, compared to literature reported values of 40-70 kJ/mol for CeO₂ supported Pt catalysts (*Science* 2013, 341, 771; *ACS Catal.* 2015, 5, 5164; *Nat. Commun.* 2019, 10:1358; *Nat. Commun.* 2019, 10:3808). Per the reviewer's suggestion, we re-evaluated the catalytic activity of Ce_{0.99}Cu_{0.01}O₂ support by calculating the reaction rates per mass of catalyst. As shown in Fig. R3, the activation energy of Ce_{0.99}Cu_{0.01}O₂ is determined to 77.60 kJ/mol, which is close to those of previous reports (*ACS Catal.* 2017, 7, 1313; *J. Am. Chem. Soc.* 2019, 141, 17548; *ACS Catal.* 2020, 10, 4692). We have compiled relevant data in the updated supplementary information.

Figure R3. Arrhenius plot of CO oxidation rate of Cu doped CeO₂ support.

Modification:

1. Added the result and discussion of the cycling tests of Pt_n/CeCu. *“The slightly shifts of CO conversion curves in cycling tests of Pt_n/CeCu (Supplementary Fig. 9) indicate the structural stability of the constructed Pt/oxide interface, which can be attributed to the gas-phase based ALD method that has minimal detrimental effect on the surface structure of oxide supports³⁵.”*

(Second paragraph in page 7 in text and Supplementary Fig. 9 in SI)

2. Added the test conditions of activity evaluation. *“50 mg of catalysts without any pre-treatments were packed ...”, “The inert quartz was used to dilute the catalysts to make sure the appropriate CO conversion.”*

(Second paragraph in page 18 in text)

3. Added the explanation on the high activation energy of Pt_n/Ce. *“The high energy barrier for Pt₅/Ce also agrees well with the large activation barrier of Pt_n/Ce in the kinetic test, which results from the strong CO poisoning effect on the subnanometric Pt clusters.”*

(First paragraph in page 13 in text)

4. Added the result and discussion of activation energy of Ce_{0.99}Cu_{0.01}O₂ support. *“The activation energy of Ce_{0.99}Cu_{0.01}O₂ support is determined to be 77.60 kJ/mol (Supplementary Fig. 15), close to that in previous studies^{39,43,44}.”*

(First paragraph in page 8 in text and Supplementary Fig. 15 in SI)

3>> The formula in the line 363 contains parameter interface corresponding to the fraction of Pt atoms at Pt-support interface, which are also called perimeter sites. This parameter is used for TOF calculation but there are no details how this parameter was estimated for each catalyst. Besides, Table S1 compares TOFs for different Pt containing catalysts measured in this work with those previously reported in the literature. Can the authors confirm that in all these works TOF parameter was estimated based on the number of above-mentioned perimeter sites and that this number was estimated in a similar way?

Author reply:

We thank the reviewer for his/her comment on our TOF calculation. We added detailed information and procedural description to clarify the calculation method of TOF in order to make a fair comparison in Table S1. The fraction of Pt atoms at Pt/CeO₂ interfaces in our TOF calculations is estimated by the scaling relation of $d^{1.9\pm0.2}$ and $d^{2.6\pm0.1}$ for perimeter and corner atoms, in accordance with previous reported literature (*Science* 2013, 341, 771). We have carefully examined the references in Table R1 and confirmed that the TOF values in Ref. 1 and Ref. 2 are calculated by the same method. However, the TOF values in some of other references, especially for supported Pt clusters and single atoms, were calculated by considering all Pt atoms in their catalysts. Therefore, we re-scaled our TOF as the activity per each of Pt atoms in our catalysts for a fair comparison.

$$\text{TOF} = \frac{\frac{P \times V}{R \times T} \times w_{\text{CO}} \times X_{\text{CO}}}{n_{\text{Pt}}}$$

As shown in Table R1, the TOF of Pt_n/CeCu for CO oxidation at 80 °C is 0.26 s⁻¹, which is one order of magnitude higher than that of atomically-dispersed Pt catalyst and rivals the best state-of-the-art Pt/CeO₂ catalyst. In the revised manuscript, we have recalculated the TOF values and revised the corresponding discussion.

Table R1. Lists of Pt average size (D_{Pt}, nm), Pt mass loading (wt%), onset temperature of CO oxidation (T_{onset}, °C), turnover frequency (TOF, s⁻¹) and activation energy (E_a, kJ/mol) of our reported catalysts and that in previous studies. Note that the TOF is tested under the temperature in parentheses.

Sample	D _{Pt} (nm)	Loading (wt%)	T _{onset} (°C)	TOF (s ⁻¹)	E _a (kJ/mol)	References
Pt _n /CeCu	0.75	1.77 wt%	8	0.26 (80)	39.49	This work
Pt _n /Ce	0.63	0.72 wt%	36	0.03 (80)	97.62	This work
Pt ₁ /CeCu	Single atom	1.51 wt%	20	0.02 (80)	56.60	This work

Pt ₁ /Ce	Single atom	0.63 wt%	100	<10 ⁻⁴ (80)	101.43	This work
Pt/CeO ₂	1.6	0.5 wt%	/	0.2 (80) ^[a]	51	Ref. 1
Pt/CeO ₂	1.2	1.5 wt%	/	0.45 (80) ^[a]	44	Ref. 2
Pt/CeO ₂	0.5	1.3 wt%	/	0.01 (50) ^[b]	70	Ref. 3
Pt/CeO ₂	2.5	0.5 wt%	/	0.60 (200) ^[b]	63.7	Ref. 4
Pt/CeO ₂	1.68	1.0 wt%	25	0.101 (80)	30.1	Ref. 5
Pt/CeO ₂	1.0	2.8 wt%	30	1.97 (150)	40	Ref. 6
Pt ₁ /CeO ₂	Single atom	1.0 wt%	100	0.005 (80)	53.5	Ref. 5
Pt ₁ /CeO ₂	Single atom	1.0 wt%	150	0.12 (225)	57	Ref. 7
Pt ₁ /CeO ₂	Single atom	1.0 wt%	60	0.08 (80)	42.5	Ref. 8
Pt ₁ /CeO ₂	Single atom	0.22 wt%	60	/	61.5	Ref. 9

*Note: [a] The TOFs are calculated based on the fraction of Pt atoms at Pt/CeO₂ interface. [b] The TOFs are normalized to the value per exposed surface Pt atoms.

Ref. 1. Cargnello, M. et al. Control of metal nanocrystal size reveals metal-support interface role for ceria catalysts. *Science* **341**, 771-773 (2013).

Ref. 2. Kopelent, R. et al. Catalytically active and spectator Ce³⁺ in ceria-supported metal catalysts. *Angew. Chem. Int. Ed.* **54**, 8728-8731 (2015).

Ref. 3. Ke, J. et al. Strong local coordination structure effects on subnanometer PtO_x clusters over CeO₂ nanowires probed by low-temperature CO oxidation. *ACS Catal.* **5**, 5164-5173 (2015).

Ref. 4. An, K. et al. Enhanced CO oxidation rates at the interface of mesoporous oxides and Pt nanoparticles. *J. Am. Chem. Soc.* **135**, 16689-16696 (2013).

Ref. 5. Pereira-Hernández, X. I. et al. Tuning Pt-CeO₂ interactions by

high-temperature vapor-phase synthesis for improved reducibility of lattice oxygen. *Nat. Commun.* 10:1358 (2019).

Ref. 6. Wang, H. et al. Surpassing the single-atom catalytic activity limit through paired Pt-O-Pt ensemble built from isolated Pt₁ atoms. *Nat. Commun.* 10:3808 (2019).

Ref. 7. Jones, J. et al. Thermally stable single-atom platinum-on-ceria catalysts via atom trapping. *Science* **353**, 150-154 (2016).

Ref. 8. Nie, L. et al. Activation of surface lattice oxygen in single-atom Pt/CeO₂ for low-temperature CO oxidation. *Science* **358**, 1419-1423 (2017).

Ref. 9. Wang, C. L. et al. Water-mediated Mars-van Krevelen mechanism for CO oxidation on ceria-supported single-atom Pt₁ catalyst. *ACS Catal.* **7**, 887-891 (2017).

Modification:

1. Revised the TOF calculation formula and the values of Pt_n/Ce and Pt_n/CeCu in Supplementary Table 1.

(First paragraph in page 19 in text and Supplementary Table 1)

2. Revised the discussion of the TOF values of our prepared catalysts. “*Cu-modified CeO₂-supported Pt sub-nanoclusters demonstrate a remarkable performance with an onset of CO oxidation reactivity below room temperature, which is one order of magnitude more active than atomically-dispersed Pt catalysts.*”

(Abstract in page 2 in text)

3. “...turnover frequency (TOF) of 0.88 s⁻¹ at 80 °C...” was revised to “...turnover frequency (TOF) of 0.26 s⁻¹ at 80 °C...”.

(Second paragraph in page 4 in text)

4. “The TOF of Pt_n/CeCu catalyst can reach to 0.88 s⁻¹ at 80 °C, which is superior to ...” was revised to “The TOF of Pt_n/CeCu catalyst reaches 0.26 s⁻¹ at 80 °C, which rivals other previously reported Pt/CeO₂ catalysts including Pt single atoms, clusters and nanoparticles (Supplementary Table 1).”.

(First paragraph in page 8 in text)

4>> The authors report (based of the DRIFT data) that CO absorption strength gets lower for Pt_n/CuCe in comparison to Pt_n/Ce catalyst. This effect is interesting and novel. It would make sense to confirm it by TPD experiments. Besides, measuring of the reaction orders for CO and oxygen would be useful, as weaker CO adsorption on Pt_n/CuCe can affect the reaction mechanism in different ways. May be reaction on Pt_n/CuCe takes place not only at the interface but also on Pt nanoparticles surface due to weaker CO adsorption?

Author reply:

We thank the reviewer's interest in our reported phenomenon about the lowered CO adsorption strength for Pt_n/CeCu. The CO adsorption strength is lower for Pt_n/CeCu than Pt_n/Ce according to our newly conducted *in situ* DRIFTS characterizations. We agree with the reviewer that in-depth analysis on this phenomenon and its effects on catalytic mechanism should be strengthened. The temperature-programed desorption of CO (CO-TPD) has been performed by the VDSorb-91x chemisorption analyzer. 100 mg of the catalyst is supported by quartz wool in a U-type quartz tube reactor. After being pretreated using 50 mL/min of He at 200 °C for 30 min, the catalyst is cooled down to -100 °C by liquid nitrogen trap under He flow. The catalyst is exposed to 50 mL/min of 10 vol. % CO balanced by He for 30 min, then the feed is switched to He to purge the catalyst until the baseline is stable. The CO-TPD curves are obtained under the He flow by using the AMETEK® quadrupole mass spectrometer to monitor the signal of CO ($m/z = 28$) and CO₂ ($m/z = 44$), when the reactor is heated to 600 °C with a ramp rate of 5 °C/min. As shown in Fig. R4, the peaks at -75 and -67 °C for Pt_n/CeCu and Pt_n/Ce can be assigned to CO desorption from Cu doped CeO₂ and CeO₂ nanorod supports, respectively, which are close to that of Pt/CeO₂ in previous study (*J. Phys. Chem. 1987, 91, 3310*). The CO desorption found at -19 °C for Pt_n/CeCu catalyst along with the formation of CO₂ can be attributed to the CO oxidation reaction at the interface. The desorption temperature of CO on Pt_n/CeCu is much lower than that on Pt_n/Ce, indicating the Cu dopants can weaken the CO adsorption at the interfaces. The CO desorption and CO₂ formation at about 110 °C for Pt_n/CeCu and Pt_n/Ce can be attributed to the activation of CO

adsorbed at atop sites of Pt clusters. We have also supplemented DFT calculations of CO adsorption energies atop Pt atoms for Pt₅/CeCu and Pt₅/Ce, respectively. As shown in Fig. R5, the top Pt atoms for both Pt₅/Ce and Pt₅/CuCe will be poisoned by CO molecule with overbinding that is consistent with DRIFTS spectra in Fig. R1, i.e. some CO molecules remained after O₂ flow was introduced. Thus it can be deduced that the catalytic reaction mainly takes place at the interface of Pt_n/CeCu and Pt_n/Ce catalyst at room temperature.

Figure R4. TPD curves following a saturation adsorption of CO on (a) Pt_n/CeCu and (b) Pt_n/Ce catalysts at -100 °C.

Figure R5. Atomic structures and adsorption energies of CO adsorbed on the top Pt atoms of (a) Pt₅/Ce and (b) Pt₅/CeCu.

As to the *in situ* DRIFTS spectra of CO oxidation shown in Fig. R1, there are obvious stretching signals of CO₂ molecules for Pt_n/CeCu catalysts in CO flow,

indicating that the CO molecules are reacted away by active oxygens in the support. After the flow is switched to He flow, stretching signals of CO₂ molecules decreased, which are related to the decreased CO adsorption signals. When O₂ flow is introduced into the DRIFTS cell, the stretching signals of CO₂ molecules reappeared, implying that the supplement of O₂ source can further react with the adsorbed CO molecules. Therefore, we believe that the loss of adsorbed CO molecule can be attributed to the oxidation reaction of weakly bound CO with active oxygens. The reaction orders of CO and O₂ over Pt_n/CeCu have also been investigated, with both of them close to zero as shown in Fig. R6, indicating a negligible competitive adsorption between CO and O₂ during CO oxidation over Pt_n/CeCu. These results imply that the catalytic reaction path follows the Mars-van Krevelen mechanism involving the interfacial sites, which agrees well with our DFT calculations and recent literatures on Pt/CeO₂ catalysts (*Nat. Commun.* 2019, 10: 1358; *ChemCatChem* 2020, 12, 1726). In the revised manuscript, we have added the data and corresponding discussion to strengthen our catalytic mechanism study.

Figure R6. The reaction orders of CO and O₂ over Pt_n/CeCu.

Modification:

1. Added the result and discussion of CO-TPD. “The temperature-programmed

desorption of CO has also been performed to investigate the desorption behavior of CO on Pt_n/CeCu and Pt_n/Ce (Supplementary Fig. 21). The CO desorption found at -19 °C for Pt_n/CeCu catalyst along with the formation of CO₂ can be attributed to the CO oxidation reaction at interfaces. The desorption temperature of CO on Pt_n/CeCu is much lower than that on Pt_n/Ce, indicating the weakening of interfacial CO adsorption by Cu dopants.”

(First paragraph in page 10 in text and Supplementary Fig. 21 in SI)

2. Revised the discussion of *in-situ* DRIFTS results of Pt_n/CeCu. “*On the other hand, in situ DRIFTS spectra of Pt_n/CeCu in Fig. 2d clearly show the loss of adsorbed CO molecules under He flow after CO exposure at the room temperature. The stretching signals of formed CO₂ molecules under CO flow also decrease, which are related to the decreased CO adsorption signals (Supplementary Fig. 19)...*” “*Thus, the interfacial oxygen site of Pt_n/CeCu is activated and CO adsorption strength at the interface is weakened,...*”

(Second paragraph in page 9 in text)

3. Added the result and discussion of reaction order test. “*The reaction orders of CO and O₂ over Pt_n/CeCu are close to zero (Supplementary Fig. 14), which indicates the negligible competitive adsorption between CO and O₂ during CO oxidation at the interfaces of Pt_n/CeCu.*”

(First paragraph in page 8 in text and Supplementary Fig. 14 in SI)

4. Added the result and discussion of CO adsorption on the top Pt atoms of Pt₅/Ce and Pt₅/CeCu. “*The adsorption energies of CO on the supported Pt single atom and cluster have been calculated as shown in Supplementary Fig. 26. The top Pt atoms for both Pt₅/Ce and Pt₅/CuCe will be poisoned by CO molecule with overbinding that is consistent with our DRIFTS results, i.e. some CO molecules remained after O₂ flow was introduced (Fig. 2d).*”

(Second paragraph in page 12 in text and Supplementary Fig. 26 in SI)

5. Added the method for CO-TPD experiments. “*The temperature-programmed desorption of CO (CO-TPD) was performed by the VDSorb-91x chemisorption analyzer. 100 mg of the catalyst was supported by quartz wool in a U-type quartz*

tube reactor. After being pretreated using 50 mL/min of He at 200 °C for 30 min, the catalyst was cooled down to -100 °C by liquid nitrogen trap under He flow. The catalyst was exposed to 50 mL/min of 10 vol. % CO balanced by He for 30 min, then the feed was switched to He to purge the catalyst until the baseline was stable. CO-TPD curves were obtained under the He flow by using the AMETEK® quadrupole mass spectrometer to monitor the signal of CO ($m/z = 28$) and CO₂ ($m/z = 44$), when the reactor was heated to 600 °C with a ramp rate of 5 °C/min.”

(Third paragraph in page 17 in text)

5>> The EXAFS analysis involving second coordination shell of Pt reported in Table S2 does not look very reliable: the fitted curves are not presented, the fitted ΔE of 21 -29 eV is too high, which suggests that fits might not be correct. Besides, basic description of XAS beamline is missing: type of monochromator, mirrors, detectors, flux, beam size...

Author reply:

We appreciate reviewer's sharp comment and agree with the reviewer that the fitted ΔE_0 of 21 ~ 29 eV is too high and the fitting of second coordination shell of Pt needs some reconsideration. Previously, the first and second shells in the EXAFS data of Pt_n/Ce and Pt_n/CeCu have been fitted with Pt-O and Pt-Pt paths, respectively. We tried refitting the Pt k^3 -weighted Fourier transformed EXAFS data of Pt_n/Ce and Pt_n/CeCu. The fitting curves and the contribution of respective backscattering paths are presented in Fig. R7. The fitted ΔE_0 of Pt-Pt paths for Pt_n/Ce and Pt_n/CeCu are determined to 14.68 ± 8.63 and 16.84 ± 9.77 , respectively, which are still too high with relatively large error. Considering the low crystallinity and high disorder of Pt atoms for the deposited Pt clusters at low temperature, the Pt-Pt contribution in the EXAFS data of Pt_n/Ce and Pt_n/CeCu is very small, which have also been mentioned in previous literatures (*Nat. Chem.* 2011, 3, 634; *Angew. Chem. Int. Ed.* 2017, 56, 13078; *J. Am. Chem. Soc.* 2020, 142, 169). For this reason, we have deleted the fitting results of Pt-Pt paths for Pt_n/Ce and Pt_n/CeCu in the revised manuscript. We have

presented the detailed fitting data of Pt-O paths for our samples as shown in Fig. R8, and the corresponding structural information and fitting parameters have also been updated in Table R2.

We thank the reviewer for pointing out that missing information of our used X-ray absorption spectroscopy (XAS) beamline. In the updated manuscript, we have added the corresponding information of XAS experiments in details. The incident photon beam was selected by a double-crystal Si (111) monochromator after a collimating mirror and focused by a toroidal mirror. All XAS measurements were conducted in transmission mode using a 19-element high-purity germanium solid-state detector. The X-ray beam size on our prepared catalysts was about $0.9 \times 0.3 \text{ mm}^2$ at half-maximum (FWHM) with a photon flux of $> 4 \times 10^{11}$ photons/s at 9 keV.

Figure R7. Fitting of Fourier transformed EXAFS spectra of (a) Pt_n/Ce and (b) Pt_n/CeCu .

Figure R8. Experimental (black line) and fitting (red circle) results of Fourier transformed EXAFS spectra (left: R-space and right: k-space) of (a) Pt₁/Ce, (b) Pt₁/CeCu, (c) Pt_n/Ce, (d) Pt_n/CeCu, (e) Pt foil and (f) PtO₂.

Table R2. Structural information and fitting parameters obtained from Pt L_{III}-edge EXAFS spectra of catalysts.

Sample	Shell	N	R (Å)	σ^2 (Å ²)	ΔE_0 (eV)
Pt ₁ /Ce	Pt-O	3.81 ± 0.63	2.010 ± 0.008	0.0012 ±	11.58 ±
				0.0011	1.03
Pt ₁ /CeCu	Pt-O	4.41 ± 0.28	2.004 ± 0.006	0.0010 ±	10.64 ±
				0.0007	0.74
Pt _n /Ce	Pt-O	2.89 ± 0.37	2.009 ± 0.013	0.0018 ±	10.37 ±
				0.0015	1.53
Pt _n /CeCu	Pt-O	3.06 ± 0.42	2.016 ± 0.014	0.0024 ±	12.77 ±
				0.0018	1.60
Pt foil	Pt-Pt	12	2.765 ± 0.002	0.0046 ±	9.15 ±
				0.0003	0.39
PtO ₂	Pt-Pt	6	3.088 ± 0.008	0.0030 ±	5.79 ±
				0.0008	2.20
	Pt-O	6	2.016 ± 0.006	0.0018 ±	9.95 ±

Modification:

1. Revised the fitting data in Supplementary Fig. 25 and fitting parameters in Supplementary Table 2.

(Supplementary Fig. 25 and Table 2 in SI)

2. Added the discussion of the weak signals of Pt-Pt bond for Pt_n/Ce and Pt_n/CeCu catalysts. *“The small Pt-Pt contribution can be attributed to the low crystallinity and high disorder of Pt atoms for the deposited Pt clusters at relatively low temperature as presented in the HADDF-STEM images, which agrees with previously reports^{20,41}.”*

(First paragraph in page 11 in text)

3. Added the test conditions of XAS experiments. *“The incident photon beam was selected by a double-crystal Si (111) monochromator after a collimating mirror and focused by a toroidal mirror. All XAS measurements were conducted in transmission mode using a 19-element high-purity germanium solid-state detector. The X-ray beam size on our prepared catalysts was about 0.9 × 0.3 mm² at half-maximum (FWHM) with a photon flux of > 4 × 10¹¹ photons/s at 9 keV.”*

(First paragraph in page 18 in text)

6>> Figure 3c compares the calculated Bader charges of interfacial Pt atoms to the structural parameters of Pt determined by XAS measured ex situ. What is the origin of these correlations from the theoretical point of view? Can the authors add the error bars to the plots to confirm that the changes are significant? How relevant are the reported correlations considering that in air most of Pt atoms in Pt_n particles are oxidized by air, while during catalysis they should be partially reduced, thus large fraction of oxygen in their local coordination should be replaced by CO?

Author reply:

We thank the reviewer for his/her constructive comments. Our proposed calculation

models are based on our analysis of the electronic and structural information of the XAS results. The Bader analysis has usually been performed to obtain the atomic charges and study the interfacial electronic interactions (*Comput. Mater. Sci.*, 2006, 36, 354), which can be directly related to the white line intensity that reflects the oxidation state of Pt (*J. Synchrotron Radiat.* 1999, 6, 471). As shown in Fig. R9, the error bars of fitted bond length and coordinate number from EXAFS data has been added, which show the same trend as the structural parameters from our constructed models. In fact, in order to avoid the influence of air during XAS experiments, all catalysts have been tested under N₂ flow at room temperature in the cell shown in Fig. R10 that has been reported in our previous study (*Angew. Chem. Int. Ed.* 2017, 56, 1648). We have described the XAS experiments conditions in details in the updated manuscript.

Figure R9. Q_{Pt}, calculated Bader charges of interfacial Pt atoms; L_{Pt-O}, bond lengths between Pt atoms and interfacial oxygen; and CN_{Pt-O}, coordinate number of interfacial Pt atoms, of Pt₁/CeCu, Pt₁/Ce, Pt_n/CeCu and Pt_n/Ce. The black square points are the white line intensity from XANES spectra. The red and magenta square points representing L_{Pt-O} and CN_{Pt-O} are the fits to the EXAFS spectra.

Figure R10. The cell for XAS experiments. The XAS results of the catalysts are collected under N_2 flow at room temperature without any pretreatments.

Modification:

1. Added the explanation of Bader analysis. “*the calculated Bader charges⁴⁷ of interfacial Pt atoms (Q_{Pt}) show the same trend as the white line intensity that are related to electron transfer at the interface.*”

(First paragraph in page 12 in text)

2. Added error bars of L_{Pt-O} and CN_{Pt-O} in Fig. 3c.

(Fig. 3c in text)

3. Added the test conditions of XAS experiments. “*In order to avoid the influence of air, all catalysts were tested under N_2 flow at room temperature in the cell as our previous study reported³⁴.*”

(First paragraph in page 18 in text)

7>> Concerning XPS measurements, were they performed in vacuum? Was the possibility of photo-reduction of Ce^{4+} during the measurements considered?

Author reply:

The XPS characterizations of all samples are performed in high vacuum environment

($\sim 10^{-7}$ Pa). All samples are kept in the XPS chamber overnight before the acquisition. The acquisition time is 30 min. We agree with the reviewer that the high-vacuum XPS characterization can overestimate the concentration of Ce^{3+} due to the photo-reduction of Ce^{4+} , which has been reported in previous studies (*Surf. Sci.* 2004, 563, 74; *Langmuir* 1996, 12, 1794). However, despite this overestimation, XPS is still a useful tool to qualitatively investigate the surface oxygen activity of our catalysts, especially in conjunction with other techniques such as Raman spectra. This kind of combined characterization technique has been widely used to investigate the surface properties of CeO_2 based catalysts in many previous literature studies (*Science* 2007, 318, 1757; *Nat. Commun.* 2019, 10:1358; *Nat. Commun.* 2019, 10:3808). In our study, both the XPS and Raman results agree well with the formation energies of surface and interfacial oxygen vacancy from our DFT calculations. Moreover, the temperature-programmed reduction by hydrogen (H_2 -TPR) has been performed to investigate the surface reducibility of our prepared catalysts as shown in Fig. R11. The reduction peaks below 200 °C are assigned to the reduction of oxidized Pt species. Cu dopants cause the shift of the reduction peaks of surface oxygen (250 ~ 550 °C) and bulk oxygen (about 710 °C) of CeO_2 supports for Pt_1/Ce and Pt_n/Ce to low temperatures. The reducibility of surface oxygen follows the sequences of $\text{Pt}_n/\text{CeCu} > \text{Pt}_1/\text{CeCu} > \text{Pt}_n/\text{Ce} > \text{Pt}_1/\text{Ce}$, which agrees well with our XPS results. In our updated manuscript, we have strengthened the discussion on the activity of surface oxygen.

Figure R11. H₂-TPR profiles of Pt_n/CeCu, Pt_n/Ce, Pt₁/CeCu and Pt₁/Ce catalysts.

Modification:

1. Added the result and discussion of H₂-TPR. *“The temperature-programmed reduction by hydrogen (H₂-TPR) has also been performed to investigate the surface reducibility of our prepared catalysts (Supplementary Fig. 18). Cu dopants also cause the shift of reduction peaks of surface oxygen (250 ~ 550 °C) and bulk oxygen (about 710 °C) of CeO₂ supports for Pt₁/Ce and Pt_n/Ce to low temperatures. The reducibility of surface oxygen follows the sequences of Pt_n/CeCu > Pt₁/CeCu > Pt_n/Ce > Pt₁/Ce, indicating that both Cu doping and change of Pt size can affect the surface reducibility of our prepared catalysts.”*

(First paragraph in page 9 in text and Supplementary Fig. 18 in text)

2. Added the discussion of surface oxygen vacancy by combining the experimental and theoretical results. *“The formation energies of O_v at the interface of our constructed models have been presented in Supplementary Fig. 27.”*

(Second paragraph in page 12 in text)

3. Added the test condition of XPS characterization. *“...in high vacuum environment (~10⁻⁷ Pa) after all samples are kept in the XPS chamber overnight...”*

(Second paragraph in page 16 in text)

8>> *The term "modulation" used in the manuscript title seems to be not exact for the reported phenomena.*

Author reply:

We thank the reviewer for this suggestion. The term "modulation" emphasizes on the regularity of the results with changing parameters, which might not be suitable in this case. The key to activate the catalytic activity of supported Pt catalysts in our work is the precise control of interfacial reducibility and structure via oxide doping and redox-coupled atomic layer deposition process. To highlight the importance of this fabrication strategy and its beneficial effects, we have changed the title of our manuscript to "*Activation of subnanometric Pt on Cu-modified CeO₂ via redox-coupled atomic layer deposition for low-temperature CO oxidation*".

Modification:

1. The title of our manuscript has been revised to "*Activation of subnanometric Pt on Cu-modified CeO₂ via redox-coupled atomic layer deposition for low-temperature CO oxidation*".

(Title in Page 1)

Reviewer #2:

The authors have studied CO oxidation on Pt deposited on Cu doped ceria nanorods. The results show high reactivity for Pt sub-nano clusters for CO oxidation. There is considerable interest in improving the reactivity of Pt catalysts for this reaction, so the work is potentially interesting. However, the authors have overlooked some recent literature, which causes me to question their interpretation. The approach they used to report reactivity, TOF based on interfacial sites, is not consistent with standard practice in this field. Finally, they state that Cu doped ceria is not active for CO oxidation at low temperatures, which is not true. For these reasons, I do not think the manuscript is not suitable for publication in its present form.

Author reply:

We appreciate the reviewer for his/her positive evaluation on our work and pointing us to some excellent works in this field. We have carefully read these references and improved the discussion of the catalytic mechanism part in our study. We have revised the corresponding discussion including the activity of Cu doped ceria and the catalytic mechanism, as well as the evaluation and comparison of TOF in the updated manuscript. We hope the updated manuscript will deem fit for the publication in *Nature Communications*. Below is our point by point response to reviewer comments.

I>> Cu doped ceria is known to be active for CO oxidation as seen in reference 1 and 2 below. Atomically dispersed Cu shows onset of CO oxidation reactivity starting at room temperature.

1. W.-Z. Yu, W.-W. Wang, S.-Q. Li, X.-P. Fu, X. Wang, K. Wu, R. Si, C. Ma, C.-J. Jia, and C.-H. Yan, Construction of Active Site in a Sintered Copper–Cerium Nanorod Catalyst Journal of the American Chemical Society 141 (44) (2019) 17548-17557 DOI: 10.1021/jacs.9b05419.

2. K. Kappis, C. Papadopoulos, J. Papavasiliou, J. Vakros, Y. Georgiou, Y. Deligiannakis, and G. Avgouropoulos, Tuning the Catalytic Properties of

DOI:

Author reply:

We thank the reviewer for pointing us to relevant excellent works in this field. We are aware that Cu doped ceria catalysts can be activated at room temperature for CO oxidation by controlling the mass loading, distribution and chemical state of Cu species, as well as the morphology, specific surface area and interface structure of catalysts. (*J. Am. Chem. Soc.* 2019, 141, 17548; *Catalysts* 2019, 9, 138; *ACS Catal.* 2017, 7, 1313; *ACS Catal.* 2017, 7, 6843). However, it is generally accepted and consistent with aforementioned references, that the concentration of Cu should be larger than 1 wt% to achieve the excellent low-temperature activity. The concentration of Cu dopants in our Pt_n/CeCu is only 0.20 wt% per our ICP-OES characterization. We expect that such low concentration of Cu dopants would not induce the formation of CuO_x species, but rather to control the surface reducibility of CeO₂ support. With the same Cu concentration, the activity of our Ce_{0.99}Cu_{0.01}O₂ support is similar to that of CuCe-120-0.05 sample in reference 2 (*Catalysts* 2019, 9, 138). Therefore, the low activity of our Cu doped ceria at low temperature is due to the low concentration of Cu dopants, whose catalytic behavior is quite different from active Cu doped ceria. We agree with the reviewer that our expression of the essentially inactive Cu doped ceria is inappropriate. We have revised the corresponding description and cited the related references in the updated manuscript.

Modification:

1. Revised the description of the activity of our prepared Cu doped CeO₂ catalyst. “*The concentration of Cu dopants is kept low in Ce_{0.99}Cu_{0.01}O₂ to avoid the formation of CuO_x species, and catalytic testing show that the contribution of our Ce_{0.99}Cu_{0.01}O₂ support to reactivity below 100 °C is negligible (Supplementary Fig. 10), in agreement with the comparable activity of Cu doped CeO₂ catalyst in previous study⁴².*”

(Second paragraph in page 7 in text)

2>> The authors state that the adsorbed CO is lost when flowing He after CO oxidation (Figure 2d). A similar observation was reported in reference 3 where the sample does not contain any Cu. Furthermore, reference 4 shows clearly that this loss of adsorbed CO is a result of oxygen being supplied to the Pt clusters from the ceria support. It is not a result of weakening of the metal-CO bond. Hence, the authors need to consider these references and revise their explanation.

3. X.I. Prcira-Hernandez, A. DeLaRiva, V. Muravev, D. Kunwar, H. Xiong, B. Sudduth, M. Engelhard, L. Kovarik, E.J.M. Hensen, Y. Wang, and A.K. Datye, Tuning Pt-CeO₂ interactions by high-temperature vapor-phase synthesis for improved reducibility of lattice oxygen *Nature Communications* 10 (2019) DOI: 10.1038/s41467-019-09308-5.

4. Y. Lu, C. Thompson, D. Kunwar, A.K. Datye, and A.M. Karim, Origin of the High CO Oxidation Activity on CeO₂ Supported Pt Nanoparticles: Weaker Binding of CO or Facile Oxygen Transfer from the Support? *Chemcatchem* (2020) DOI: 10.1002/cctc.201901848.

Author reply:

We truly appreciate the reviewer for his/her constructive comment and providing us helpful references about catalytic mechanism studies on Pt/CeO₂ catalysts. We have performed the *in situ* DRIFTS experiments of CO oxidation of Pt_n/CeCu catalyst as shown in Fig. R12. There are obvious stretching signals of CO₂ molecules at 2360 cm⁻¹ and 2330 cm⁻¹ for Pt_n/CeCu catalysts in CO flow, indicating that the CO molecules are reacted away by active oxygens in support. After the flow is switched to He flow, the stretching signals of CO₂ molecules decrease, which are related to the decreased CO adsorption signals. Moreover, the signals of adsorbed CO molecules decrease when the flow is switched O₂ flow, suggesting that the adsorbed CO molecules can be further oxidized by activated O₂ molecules at the interface. Therefore, we believe that the loss of adsorbed CO molecule can be attributed to the oxidation reaction with active oxygens, which is caused by the weak CO adsorption

strength and active interface sites. This result agrees well with our DFT calculations and two relevant references (*Nat. Commun.* 2019, 10: 1358; *ChemCatChem* 2020, 12, 1726). In the revised manuscript, we have cited the two references and added the corresponding discussion of *in situ* DRIFTS result to make the catalytic mechanism studies of our prepared catalysts more clearly.

Figure R12. *In situ* DRIFTS spectra of CO adsorption and oxidation of Pt_n/CeCu. After CO exposure, He flow is continued and the spectra are recorded at 2, 4, 6, 8, 10 min. Subsequently, the flow is switched to 1% vol. O₂ balanced by N₂ and the spectrum is recorded at 2 min.

Modification:

1. Added the result and discussion of *in-situ* DRIFTS spectra of CO oxidation of Pt_n/CeCu. “On the other hand, *in situ* DRIFTS spectra of Pt_n/CeCu in Fig. 2d clearly show the loss of adsorbed CO molecules under He flow after CO exposure at the room temperature. The stretching signals of formed CO₂ molecules under CO flow also decrease, which are related to the decreased CO adsorption signals (Supplementary Fig. 19). When the flow is switched from He to O₂, the signal of adsorbed CO molecules is further weakened. The appearance of stretching bands of

CO₂ molecule at 2360 cm⁻¹ and 2330 cm⁻¹ suggests that the adsorbed CO molecules on Pt_n/CeCu can be further oxidized by activated O₂ molecules at the interface, which is typical in Pt/CeO₂ catalysts^{45,46}. ... Thus, the interfacial oxygen site of Pt_n/CeCu is activated and CO adsorption strength at the interface is weakened, ...”

(Second paragraph in page 9 in text)

2. Revised Fig. 2d and Supplementary Fig. 18

(Fig. 2d in text and Supplementary Fig. 18 in SI)

3>> They report a turnover frequency calculated using a term w interface (line 363). This is the ratio of active Pt atoms to total Pt atoms. It is not clear how this number is calculated. It is notoriously difficult to pinpoint the number of interface sites, if those are the only ones active. Even reference 16 by Cargnello et al. who focused on the role of interfacial sites based their reactivity on total Pt atoms, and only used a model to make their case that corner atoms are likely to be active. The standard practice in the literature on single atom catalysts is to simply report the TOF based on total Pt atoms. Hence the authors need to use the more conventional definition and then compare their reactivity with state of the art Pt-ceria catalysts, such as those in reference 3 above or reference 5 below.

5. H. Wang, J.-X. Liu, L.F. Allard, S. Lee, J. Liu, H. Li, J. Wang, J. Wang, S.H. Oh, W. Li, M. Flytzani-Stephanopoulos, M. Shen, B.R. Goldsmith, and M. Yang, Surpassing the single-atom catalytic activity limit through paired Pt-O-Pt ensemble built from isolated Pt-I atoms Nature Communications 10 (2019) DOI: 10.1038/s41467-019-11856-9.

Author reply:

We thank the reviewer for his/her comment on our TOF evaluation and agree with the reviewer that TOF calculated by total Pt atoms is more appropriate for a fair comparison with other supported Pt catalysts, especially for Pt single atoms and clusters. We have redefined our TOF as the activity per each of Pt atoms in our catalysts.

$$\text{TOF} = \frac{\frac{P \times V}{R \times T} \times w_{\text{CO}} \times X_{\text{CO}}}{n_{\text{Pt}}}$$

As shown in Table R3, the TOF of Pt_n/CeCu for CO oxidation at 80 °C is 0.26 s⁻¹, which is one order of magnitude higher than that of atomically-dispersed Pt catalyst. Note that the TOF values for reference 9-12 in Table S1 in supplementary information are calculated based on the fraction of interfacial or surface Pt atoms. Therefore, the catalytic activity of Pt_n/CeCu still rivals state-of-the-art Pt/CeO₂ catalysts for low-temperature CO oxidation. In the updated manuscript, we have recalculated the TOF values and revised the corresponding discussion.

Table R3. Lists of Pt average size (D_{Pt}, nm), Pt mass loading (wt%), onset temperature of CO oxidation (T_{onset}, °C), turnover frequency (TOF, s⁻¹) and activation energy (E_a, kJ/mol) of our reported catalysts and that in previous studies. Note that the TOF is tested under the temperature in parentheses.

Sample	D _{Pt} (nm)	Loading (wt%)	T _{onset} (°C)	TOF (s ⁻¹)	E _a (kJ/mol)	References
Pt _n /CeCu	0.75	1.77 wt%	8	0.26 (80)	39.49	This work
Pt _n /Ce	0.63	0.72 wt%	36	0.03 (80)	97.62	This work
Pt ₁ /CeCu	Single atom	1.51 wt%	20	0.02 (80)	56.60	This work
Pt ₁ /Ce	Single atom	0.63 wt%	100	<10 ⁻⁴ (80)	101.43	This work

Modification:

1. Revised the TOF calculation formula and the values of Pt_n/Ce and Pt_n/CeCu in Supplementary Table 1.

(First paragraph in page 19 in text and Supplementary Table 1)

2. Revised the discussion of the TOF values of our prepared catalysts. “*Cu-modified*

CeO₂-supported Pt sub-nanoclusters demonstrate a remarkable performance with an onset of CO oxidation reactivity below room temperature, which is one order of magnitude more active than atomically-dispersed Pt catalysts.”

(Abstract in page 2 in text)

3. “...turnover frequency (TOF) of 0.88 s⁻¹ at 80 °C...” was revised to “...turnover frequency (TOF) of 0.26 s⁻¹ at 80 °C...”.

(Second paragraph in page 4 in text)

4. “The TOF of Pt_n/CeCu catalyst can reach to 0.88 s⁻¹ at 80 °C, which is superior to ...” was revised to “The TOF of Pt_n/CeCu catalyst reaches 0.26 s⁻¹ at 80 °C, which rivals other previously reported Pt/CeO₂ catalysts including Pt single atoms, clusters and nanoparticles (Supplementary Table 1).”.

(First paragraph in page 8 in text)

Reviewer #3:

This paper proposes a redox-coupled ALD method to regulate oxide doping and Pt cluster size of Pt/CeCu catalysts. The synthesis might be interesting. However, subnanometric Pt anchored on CeO₂ with the best catalytic activity for CO oxidation has been widely reported in previous reports, such as ACS Catalysis 2015, 5, 5164-5173. Also, Cu-doped CeO₂ itself is also a good catalyst for low temperature of CO oxidation. In this case, the novelty of this work is not competitive for Nature Communications. In the catalytic mechanism, the in-situ experiments are expected to monitor the evolution of cerium, copper, platinum as well as oxygen vacancy. Considering the very small size of Pt cluster and the strong metal-support interaction between Pt cluster and Cu-doped CeO₂, the dynamic changes of their chemical environments are important to understand the catalytic mechanism. The current evidences are not convincing enough. I cannot suggest the publish of this work. For other comments:

Author reply:

We truly thank the reviewer for taking his/her time to evaluate our work and appreciate the reviewer's comments that help us improve the quality of our study. We report a strategy to activate the low-temperature performance of Pt catalysts on Cu-modified CeO₂ supports by a new method based on redox-coupled atomic layer deposition. The interfacial reducibility and structure of composite catalysts have been precisely tuned by oxide doping and accurate control of Pt size. We are aware that highly efficient Pt subnanometric catalysts supported on CeO₂ have been reported in previous studies (*ACS Catalysis* 2015, 5, 5164; *Nat. Commun.* 2019, 10:3808). Nevertheless, the achievement of precisely controlling interfacial structures for noble metal/oxide catalysts is still of great challenge. In this work, the Cu dopants are introduced to modulate the surface reducibility of CeO₂ support and the formed Cu-O-Ce sites are the origin of the activated oxygen for catalytic reaction. The concentration of Cu is low enough to avoid the formation of CuO_x species that are unfavorable for the activity enhancement of supported Pt catalysts as shown in Supplementary Fig. 12 and 13. On the other hand, ALD method that based on the

alternate gas-phase chemical reaction enjoys the advantage of minimal destruction of the surface structure of oxide supports as reported in our previous works (*Chem. Sci.* 2018, 9, 2469; *Angew. Chem. Int. Ed.* 2017, 56, 1648). The size of deposited Pt catalysts in the range of sub-nanometer clusters to single atoms can be well controlled by our proposed ALD recipe, which can directly affect the coordination environment of interfacial Pt atoms that are the keys to the activity enhancement as reported in previous studies (*ACS Catalysis* 2015, 5, 5164; *Nat. Commun.* 2019, 10:3808). Moreover, the TOF of our prepared Pt_n/CeCu catalyst is higher than that reported in previous references (*ACS Catalysis* 2015, 5, 5164) and rivals the best state-of-the-art Pt/CeO₂ catalysts. We believe this method could be as well applied to other noble metal/oxide combinations and is of general interest to the broad catalysis community. We have strengthened the discussion of precise interface control and structure-activity relationship in our revised manuscript and hope the revised manuscript is appropriate for publication in *Nature Communications*.

We agree with the reviewer that *in-situ* experiments such as the *in-situ* X-ray absorption spectroscopy (XAS) experiment as reported in the previous reference (*ACS Catalysis* 2015, 5, 5164) are of great significance to study the catalytic mechanism, which can monitor the change of chemical states and local structures of active sites. Unfortunately, the *in-situ* XAS experiments could not be done due to the travel ban and the shut down/reduced operation of the beam center. To address the referee's concern, we conducted *in-situ* DRIFTS, H₂-TPR and CO-TPD that can also shed light on the change of local chemical environments in the reaction. Fig. R13 presents our *in-situ* DRIFTS results of CO oxidation of Pt_n/CeCu catalyst to further investigate the catalytic mechanism. The signals of adsorbed CO molecules decrease when the flow is switched from He to O₂. Meanwhile, the appearance of stretching bands of CO₂ molecule at 2360 cm⁻¹ and 2330 cm⁻¹ suggests that the adsorbed CO molecules on Pt_n/CeCu can be further oxidized by introduced O₂ molecules that are activated at the interface. The results agree well with the analysis of H₂-TPR and CO-TPD performed as the suggestions of reviewers in Comment 3 and 4, which are also consistent with our theoretical results. We also discover that the supported Pt single

atoms with larger coordinate number of Pt-O show lower activity than Pt sub-nanometer clusters with smaller coordinate number of Pt-O, which agrees well with the previous reference (*ACS Catalysis* 2015, 5, 5164). In our revised manuscript, we have cited this reference and strengthen our discussion of catalytic mechanism.

Figure R13. *In situ* DRIFTS spectra of CO adsorption and oxidation of Pt_n/CeCu. After CO exposure, He flow is continued and the spectra are recorded at 2, 4, 6, 8, 10 min. Subsequently, the flow is switched to 1% vol. O₂ balanced by N₂ and the spectrum is recorded at 2 min.

Modification:

1. Revised the introduction of our work in abstract. “Here we report a strategy to activate the low-temperature performance of Pt catalysts on Cu-modified CeO₂ supports by a new method based on redox-coupled atomic layer deposition. The interfacial reducibility and structure of composite catalysts was precisely tuned by oxide doping and accurate control of Pt size.”

(Abstract in page 2 in text)

2. Added the introduction of previously reported studies on CeO₂ supported highly

efficient Pt subnanometric catalysts. *“In order to further improve the atomic efficiency, the highly efficient Pt subnanometric catalysts supported on CeO₂ have been reported, which have shown enhanced low-temperature activity compared with atomically dispersed Pt catalysts.”*^{29,30}

(First paragraph in page 4 in text)

3. Added the discussion of the activity of Ce_{0.99}Cu_{0.01}O₂ with much low Cu concentration. *“The concentration of Cu dopants is kept low in Ce_{0.99}Cu_{0.01}O₂ to avoid the formation of CuO_x species, and catalytic testing show that the contribution of our Ce_{0.99}Cu_{0.01}O₂ support to reactivity below 100 °C is negligible (Supplementary Fig. 10), in agreement with the comparable activity of Cu doped CeO₂ catalyst in previous study⁴².”*

(Second paragraph in page 7 in text)

4. Added the discussion about the effect of coordinate number of Pt-O on the catalytic activity. *“The relatively small CN_{Pt-O} values of Pt_n/Ce and Pt_n/CeCu can be related to the highly catalytic activity compared with Pt₁/Ce and Pt₁/CeCu, respectively²⁹.”*

(First paragraph in page 12 in text)

5. Added the discussion of *in-situ* DRIFTS of CO oxidation of Pt_n/CeCu. *“When the flow is switched from He to O₂, the signal of adsorbed CO molecules is further weakened. The appearance of stretching bands of CO₂ molecule at 2360 cm⁻¹ and 2330 cm⁻¹ suggests that the adsorbed CO molecules on Pt_n/CeCu can be further oxidized by activated O₂ molecules at the interface, which agrees well with the previous studies on Pt/CeO₂ catalysts^{44,45}.”*

(First paragraph in page 10 in text)

6. Revised the discussion part to emphasize the novelty of our manuscript. *“Cu dopants have been introduced to modulate the surface reducibility of CeO₂ support, which lead to the formation of Cu²⁺ ions in the CeO₂ lattice and help create active Cu-O-Ce sites for tuning the interfacial structures and anchoring the Pt catalysts. ... The resulting Pt sub-nanoclusters on Cu-doped CeO₂ exhibit excellent activity with an onset of CO oxidation reactivity below room temperature.”*

(First paragraph in page 14 in text)

I>> Chemical status of Cu, in the lattice of CeO₂ or as CuO_x clusters? Also provide the evidences. Then, how about the spatial distribution of Pt cluster on CeO₂ and Cu-CeO₂?

Author reply:

We thank the reviewer's question on the chemical state of Cu and the spatial distribution of Pt cluster, which are important aspects of the catalytic mechanism. We have performed XPS characterization to study the chemical state of Cu. No appreciable signal in the Cu 2p spectra has been detected for both Pt₁/CeCu and Pt_n/CeCu samples as shown in Fig. R14, implying the low concentration of Cu on the oxide supports. The Cu K-edge XANES spectrum of Pt₁/CeCu has also been collected by using transmission mode, which indicates that Cu is in Cu²⁺ state and atomically dispersed in ceria support agreeing well with previous studies (*J. Am. Chem. Soc.* 2019, 141, 17548; *ACS Catal.* 2018, 8, 7113). Moreover, the temperature-programmed reduction by hydrogen (H₂-TPR) have been performed to investigate the redox property of Cu modified CeO₂ supports. As shown in Fig. R15, the reduction peaks at the range of 300 ~ 550 °C can be assigned to the reduction of surface oxygen, while the peak at 716 °C is attributed to the reduction of bulk oxygen of CeO₂ supports as our previous study reported (*Catal. Sci. Technol.* 2017, 7, 4462). The introducing of Cu causes these reduction peaks shifting to low temperature and the appearance of the reduction peak at 242 °C can be assigned to the reduction of Cu-O-Ce species, which is different to the reduction of highly dispersed CuO_x species with the reduction peak below 200 °C. (*J. Am. Chem. Soc.* 2019, 141, 17548; *ACS Catal.* 2017, 7, 1313). According to the XPS, XAFS and H₂-TPR results, Cu dopants are in the lattice of CeO₂, consistent to our DRIFTS results (Supplementary Fig. 13) showing no CO adsorption signals on the Ce_{0.99}Cu_{0.01}O₂ support. In terms to the spatial distribution of Pt clusters, the representative HAADF-STEM images of Pt_n/Ce and Pt_n/CeCu are presented in Fig. R16. The average distances of Pt clusters are 4.94 ± 1.21 nm and 4.60 ± 1.51 nm for Pt_n/Ce and Pt_n/CeCu, respectively.

Therefore, we can conclude that the spatial distribution of Pt clusters on CeO₂ and Cu doped CeO₂ are similar. We have added these results and corresponding discussion in our revised manuscript.

Figure R14. (a) Cu 2p XPS spectra of Pt₁/CeCu and Pt_n/CeCu. (b) Normalized Cu K-edge XANES spectrum of Pt₁/CeCu in comparison to Cu foil, Cu₂O and CuO references.

Figure R15. H₂-TPR profiles of CeO₂ and Cu doped CeO₂ supports.

Figure R16. Representative HAADF-STEM images of (a) Pt_n/Ce and (b) Pt_n/CeCu. The average distances of Pt clusters in a ceria nanorod are labelled.

Modification:

1. Added the discussion of the chemical status of Cu dopants. *“The Cu K-edge XANES spectrum of Pt₁/CeCu (Supplementary Fig. 3) shows that Cu dopants in Cu²⁺ state are atomically dispersed in ceria support, in good agreement with previous studies^{39,40}. Meanwhile, the absence of signals in the Cu 2p spectra indicates that no segregated CuO_x species are formed in CeO₂ supports due to the low Cu concentration (0.20 wt% determined by ICP-OES).”*

(First paragraph in page 6 in text and Supplementary Fig. 3 in SI)

2. Added the result of H₂-TPR of CeO₂ and Cu doped CeO₂. *“The appearance of the reduction peak at 242 °C for Ce_{0.99}Cu_{0.01}O₂ support can be assigned to the reduction of Cu-O-Ce species^{39,43}.”*

(First paragraph in page 9 in text and Supplementary Fig. 18 in SI)

3. Added the result of spatial distribution of Pt cluster for Pt_n/Ce and Pt_n/CeCu. *“The average distances of Pt clusters in a ceria nanorod are 4.94 ± 1.21 and 4.60 ± 1.51 nm for Pt_n/Ce and Pt_n/CeCu, respectively (Supplementary Fig. 6).”*

(Second paragraph in page 6 in text and Supplementary Fig. 6 in SI)

2>> *In the synthesis, can the Pt mass loading of Pt/CeCu be controlled?*

Author reply:

We appreciate the reviewer's question on the controllability of our preparation method for supported Pt catalysts. ALD is a self-limiting process that can precisely control the adsorption and nucleation of gas-phase precursors by tuning the ALD recipes, such as reactor temperature, plus pressure and time. For example, the Pt mass loading can be increased/decreased by carrying out more/fewer cycles of ALD. In this particular study, three samples of Pt_n/CeCu were prepared by the same atomic layer deposition process and we have performed ICP-OES tests to determine their Pt's mass loading. The tested mass loadings of Pt are 1.65 wt%, 1.79 wt% and 1.86 wt%, respectively, which correspond to the average Pt's mass loading of 1.77 ± 0.11 wt% with the relative error percentage of about 6%. The well-controlled mass loading of Pt can be attributed our specially designed fluidized atomic layer deposition chamber and the controllable recipes as described in our previous study (*Angew. Chem. Int. Ed.* 2017, 56, 1648). In the revised supplementary information, we have updated the Pt's mass loading of Pt_n/CeCu by the tested average value in Supplementary Table 1.

Modification:

1. Revised the mass loading of Pt_n/CeCu by the tested average value.

(Supplementary Table 1 in SI)

3>> The reducibility of all catalysts should be characterized by temperature-programmed reduction.

Author reply:

We agree with the reviewer that H₂ or CO TPR can directly reflect the reducibility of catalysts. To this end, H₂-TPR tests were performed for all catalysts using a chemisorption analyzer (AMI-300 series, Altamira Instrument). Specifically, 30 mg of the catalyst has been supported by quartz wool in a U-type quartz tube reactor, which is pretreated using 30 mL/min of Ar at 100 °C for 30 min. The feed is switched to 30 mL/min of 10% vol. H₂ balanced with Ar, when the catalyst is cooled down to room temperature. Then, the reactor is heated to 800 °C with a ramp rate of

5 °C/min and thermal conductivity detector is utilized to monitor the signal of H₂ consumption. As shown in Fig. R17, an obvious reduction peak below 200 °C for Pt₁/Ce and Pt₁/CeCu can be attributed to the reduction of oxidized Pt single atoms. (*Nat. Commun.* 2019, 10:3808) Both Pt_n/Ce and Pt_n/CeCu show weak reduction signals below 200 °C, implying that the reduction of oxidized Pt species during the redox ALD process. After introducing Cu dopants, the reduction peaks of surface oxygen (250 ~ 550 °C) and bulk oxygen (about 710 °C) of CeO₂ supports for Pt₁/Ce and Pt_n/Ce shift to lower temperature. Among four prepared catalysts, we can find that the reducibility of surface oxygen follows the sequence of Pt_n/CeCu > Pt₁/CeCu > Pt_n/Ce > Pt₁/Ce, which indicates that both Cu doping and change of Pt size can affect the surface reducibility of our prepared catalysts. The result is also consistent with XPS and Raman spectra and the formation energies of interfacial oxygen vacancy. In our revised manuscript, we have strengthened the discussion on the surface reducibility of our prepared catalysts by combining the experimental and theoretical results.

Figure R17. H₂-TPR profiles of Pt_n/CeCu, Pt_n/Ce, Pt₁/CeCu and Pt₁/Ce catalysts.

Modification:

1. Added the result and discussion of H₂-TPR. “The temperature-programmed

reduction by hydrogen (H₂-TPR) has also been performed to investigate the surface reducibility of our prepared catalysts (Supplementary Fig. 18). ...Cu dopants also cause the shift of reduction peaks of surface oxygen (250 ~ 550 °C) and bulk oxygen (about 710 °C) of CeO₂ supports for Pt₁/Ce and Pt_n/Ce to low temperatures. The reducibility of surface oxygen follows the sequences of Pt_n/CeCu > Pt₁/CeCu > Pt_n/Ce > Pt₁/Ce, indicating that both Cu doping and change of Pt size can affect the surface reducibility of our prepared catalysts.”

(First paragraph in page 9 in text and Supplementary Fig. 18 in text)

2. Added the discussion of surface oxygen vacancy by combining the experimental and theoretical results. *“The formation energies of O_v at the interface of our constructed models have been presented in Supplementary Fig. 27.”*

(Second paragraph in page 12 in text)

3. Added the discussion of the role of Cu dopants on surface reducibility. *“Cu dopants have been introduced to modulate the surface reducibility of CeO₂ support, which lead to the formation of Cu²⁺ ions in the CeO₂ lattice and help create active Cu-O-Ce sites for tuning the interfacial structures and anchoring the Pt catalysts.”*

(First paragraph in page 14 in text)

4. Added the description of H₂-TPR method. *“The temperature-programmed reduction by hydrogen (H₂-TPR) was performed by a chemisorption analyzer (AMI-300 series, Altamira Instrument). Typically, 30 mg of the catalyst was supported by quartz wool in a U-type quartz tube reactor, which was pretreated using 30 mL/min of Ar at 100 °C for 30 min. The feed was switched to 30 mL/min of 10% vol. H₂ balanced with Ar, when the catalyst was cooled down to room temperature. Then, the reactor was heated to 800 °C with a ramp rate of 5 °C/min and thermal conductivity detector was utilized to monitor the signal of H₂ consumption.”*

(Second paragraph in page 17 in text)

4>> In situ DRIFTS spectra, I doubt on the desorption of CO on the catalysts by He sweeping. Generally, the chemical adsorbed CO on Pt quite strong. Sweeping at room temperature might not be enough. I think the CO-temperature programmed desorption

should be correct choice to evaluate the adsorption strength of CO on various catalysts.

Author reply:

We appreciate the reviewer for his/her constructive comment and we recheck on the point that the signals of adsorbed CO in the *in-situ* DRIFTS spectra decrease under He flow. The adsorption energy of CO molecule on the vertex site of Pt cluster is usually very strong as our previous study reported (*Nanoscale* 2019, 11, 8150). However, the CO adsorption strength at the interfacial Pt site of Pt_n/CeCu is weakened due to the electron transfer from Pt atoms to oxide support. Per the suggestion of reviewer, the temperature-programed desorption of CO (CO-TPD) has been performed to investigate by the desorption behavior of CO. As shown in Fig. R18, the peaks at -75 and -67 °C for Pt_n/CeCu and Pt_n/Ce can be assigned to CO desorption from Cu doped CeO₂ and CeO₂ nanorod supports, respectively, which are closely to that of Pt/CeO₂ in previous study (*J. Phys. Chem.* 1987, 91, 3310). The CO desorption found at -19 °C for Pt_n/CeCu catalyst along with the formation of CO₂ can be attributed to the CO oxidation reaction at the interfaces. The desorption temperature of CO on Pt_n/CeCu is much lower than that on Pt_n/Ce, indicating the Cu dopants can weaken the CO adsorption at the interfaces. The CO desorption and CO₂ formation at about 110 °C for Pt_n/CeCu and Pt_n/Ce can be attributed to the activation of CO adsorbed at the top sites of Pt clusters. As the *in situ* DRIFTS experiments of CO oxidation of Pt_n/CeCu catalyst shown in Fig. R13, we can find that there are obvious stretching signals of CO₂ molecules at 2360 cm⁻¹ and 2330 cm⁻¹ for Pt_n/CeCu catalysts in CO flow, indicating that the CO molecules are reacted by the active oxygens in support at room temperature, which is consistent with the CO-TPD results. After the flow is switched to He flow, the stretching signals of CO₂ molecules decrease, which are related to the decreased CO adsorption signals. Therefore, the loss of adsorbed CO molecule can be attributed to the oxidation reaction with active oxygens, which is resulted by the weak CO adsorption strength and active interface sites. This result agrees well with our DFT calculations and the previous studies (*Nat. Commun.* 2019, 10: 1358; *ChemCatChem* 2020, 12, 1726).

We have strengthened the corresponding discussion of CO-TPD and *in situ* DRIFTS result in our updated manuscript.

Figure R18. TPD curves following a saturation adsorption of CO on (a) Pt_n/CeCu and (b) Pt_n/Ce catalysts at -100 °C.

Modification:

1. Added the result and discussion of CO-TPD. “*The temperature-programmed desorption of CO has also been performed to investigate the desorption behavior of CO on Pt_n/CeCu and Pt_n/Ce (Supplementary Fig. 21). The CO desorption found at -19 °C for Pt_n/CeCu catalyst along with the formation of CO₂ can be attributed to the CO oxidation reaction at interfaces. The desorption temperature of CO on Pt_n/CeCu is much lower than that on Pt_n/Ce, indicating the weakening of interfacial CO adsorption by Cu dopants.*”

(First paragraph in page 10 in text and Supplementary Fig. 21 in SI)

2. Added the discussion of *in-situ* DRIFTS results. “*On the other hand, in situ DRIFTS spectra of Pt_n/CeCu in Fig. 2d clearly show the loss of adsorbed CO molecules under He flow after CO exposure at the room temperature. Meanwhile, the stretching signals of formed CO₂ molecules under CO flow are also decreased, which are related to the decreased CO adsorption signals (Supplementary Fig. 18). When the flow is switched from He to O₂, the signal of adsorbed CO molecules is further weakened. The appearance of stretching bands of CO₂ molecule at 2360 cm⁻¹ and 2330 cm⁻¹ suggests that the adsorbed CO molecules on Pt_n/CeCu can be*

further oxidized by activated O₂ molecules at the interface, which agrees well with the previous studies on Pt/CeO₂ catalysts^{45,46}. ... Thus, the interfacial oxygen site of Pt_n/CeCu is activated and CO adsorption strength at the interface is weakened, ...”

(Second paragraph in page 9 in text)

3. Added the method for CO-TPD experiments. *“The temperature-programmed desorption of CO (CO-TPD) was performed by the VDSorb-91x chemisorption analyzer. 100 mg of the catalyst was supported by quartz wool in a U-type quartz tube reactor. After being pretreated using 50 mL/min of He at 200 °C for 30 min, the catalyst was cooled down to -100 °C by liquid nitrogen trap under He flow. The catalyst was exposed to 50 mL/min of 10 vol. % CO balanced by He for 30 min, then the feed was switched to He to purge the catalyst until the baseline was stable. CO-TPD curves were obtained under the He flow by using the AMETEK® quadrupole mass spectrometer to monitor the signal of CO (m/z = 28) and CO₂ (m/z = 44), when the reactor was heated to 600 °C with a ramp rate of 5 °C/min.”*

(Third paragraph in page 17 in text)

5>> In their DFT calculation, Pt5 metallic cluster was used. In the TEM images, the number of Pt atoms in cluster is over 30. To build a model over 30 might be not practical for DFT calculation. However, Pt5 is quite small cannot match well with experiments. Especially, only two layers of Pt with four Pt atom with support and one Pt atom on the top, it cannot reflect the true oxidation state of PtOx clusters. Also, the Pt in their real catalysts was identified as the oxidized Pt (X-ray absorption data). Also, I checked their experimental section, where no pre-reduction by H₂ had been mentioned. In my opinions, three layers of PtOx cluster with more Pt should be built on the top of supports for DFT calculations. In this case, I cannot trust their DFT calculations, in which the model did not match the experiments.

Author reply:

We appreciate the reviewer’s constructive comments on our DFT calculation models. As shown in Fig. R19, we have constructed a larger Pt₁₄ cluster on the (3 × 3)

supercell of CeO₂ and Cu doped CeO₂ slabs to avoid the interactions between two periodic Pt clusters (denoted as Pt₁₄/Ce and Pt₁₄/CeCu). In fact, the H₂ reduction step is included in the preparation process of Pt_n/Ce and Pt_n/CeCu catalysts based on the redox-coupled ALD method. Our XAS and XPS results show that metallic and oxidized Pt species exist in both Pt_n/Ce and Pt_n/CeCu catalysts. Since the electron transfer from supported Pt clusters to oxide supports, Pt atoms at the interface of Pt/CeO₂ are usually in the oxidized states, which have also been reported in previous studies (*Nat. Mater.* 2011, 10, 310; *ACS Catal.* 2016, 6, 6151; *J. Catal.* 2016, 344, 507). In this study, we have focused on the catalytic reactions on the interfacial sites. Therefore, we have calculated the oxygen vacancy formation energies and CO adsorption energies at the interfaces of oxide slabs supported Pt₁₄ cluster. The interfacial oxygen vacancy formation energies for Pt₁₄/Ce and Pt₁₄/CeCu are 1.93 and 0.73 eV, respectively, which show the same trend as that of Pt₅/Ce (2.30 eV) and Pt₅/CeCu (0.91 eV) in our manuscript. Moreover, the results of adsorption energies of CO on the interfacial Pt sites between supported Pt₁₄ and Pt₅ clusters are also similar. These results indicate that the interface composed by Pt clusters and oxide slab in the DFT calculations does reflect the characteristic local structures in Pt_n/Ce and Pt_n/CeCu and does not change our main conclusion despite the numerical variations. In the updated manuscript, we have added the results and discussion of the oxygen vacancy formation energy and CO adsorption energy of Pt₁₄/Ce and Pt₁₄/CeCu.

Figure R19. Atomic structures of (a) Pt₁₄/Ce and (b) Pt₁₄/CeCu with an oxygen vacancy. Atomic structures of CO adsorption at the interfacial Pt sites of (c) Pt₁₄/Ce and (d) Pt₁₄/CeCu.

Modification:

1. Added the added the comparison of oxygen vacancy formation energy and CO adsorption energy of between supported Pt₅ and Pt₁₄ clusters. *“In order to eliminate potential variations due to the size of Pt clusters used in the calculation, the O_v formation energies and CO adsorption energies at the interfaces of oxide slabs supported Pt₅ and Pt₁₄ cluster with larger size have also been calculated (Supplementary Fig. 28).”*

(Second paragraph in page 12 in text and Supplementary Fig. 28 in SI)

6>> In the Supplementary Figure 6, why is Cu so much less than Pt?

Author reply:

The molar ratio of Ce and Cu is kept as 99:1 in our work, which can determine the nominal Cu mass concentration in the Cu-modified catalysts is about 0.37 wt%. The Pt mass loading for Pt_n/CeCu tested by ICP-OES technique is about 1.77 wt%, which is much higher than that of Cu. Therefore, we think it is reasonable for the Cu and

Pt signal in EDX spectrum of Pt_n/CeCu. In fact, we have chosen the low concentration of Cu dopant to avoid the formation of CuO_x species, since segregated CuO_x species are unfavorable for the activity enhancement of supported Pt catalysts as shown in Supplementary Fig. 12 and 13. We have also performed the ICP-OES test to determine the Cu mass concentration of Pt_n/CeCu. The tested Cu mass concentration of Pt_n/CeCu is 0.20 wt% that is slightly lower than the nominal value, and we have added the result in the revised manuscript.

Modification:

1. Added the Cu mass concentration of Pt_n/CeCu. *“Meanwhile, no signals in the Cu 2p spectra indicate that no separated CuO_x species are formed on the surface of CeO₂ supports due to so low Cu concentration (0.20 wt% determined by ICP-OES).”*

(First paragraph in page 6 in text)

7>> Comparing the results of Figure 3 and Supplementary Figure 14, why was no Pt⁴⁺ find from the Pt 4f XPS analysis?

Author reply:

We appreciate the reviewer’s concern on valence states of Pt analyzed by the XPS results of our catalysts. As shown in Figure 3, the white line intensities of Pt₁/CeCu and Pt₁/Ce are close to that of PtO₂, indicating that the supported Pt single atoms are mainly in oxidized states. In the Pt 4f XPS spectra, we can find that Pt in Pt₁/CeCu and Pt₁/Ce are mainly in Pt²⁺ state with the peaks at 72.6 eV and 76.0 eV. There are only weak signals for Pt⁴⁺ at about 74.4 eV and 77.8 eV, which agrees with previously reported studies (*Science* 2017, 358, 1419; *ACS Catal.* 2017, 7, 887; *ACS Sustainable Chem. Eng.* 2018, 6, 14054). Per the theoretical and experimental work reported by Neyman et al. (*Angew. Chem. Int. Ed.* 2014, 53, 10525), the atomically dispersed Pt⁴⁺ ions on CeO₂ is unstable, and will be totally transformed into Pt²⁺ ions when the temperature increases from 110 K and 450 K.

Reviewers' Comments:

Reviewer #1:

Remarks to the Author:

The authors have successfully improved the manuscript, which in my opinion now can be published as it is.

Reviewer #2:

Remarks to the Author:

The authors have revised the manuscript in response to the reviewer comments and have addressed most of the concerns. The work shows that the presence of Cu dopants allows the Pt samples with sub-nanometer clusters to achieve a high reactivity. When the reactivity is properly normalized to total Pt atoms, the performance is comparable to the best Pt catalysts, and the work implies that oxygen activation of the support was achieved by the Cu dopants. The work is suitable for publication. I have only two revisions to suggest.

1) The authors need to check the indexing of ceria lattice planes in Figure 1. The (110) lattice plane is forbidden by symmetry rules, it is the (220) that is allowed. They need to check their calibrations and also the planes allowed by symmetry considerations. Furthermore, it has been shown in the literature that ceria rods expose (111) and (100) surfaces [1]. The authors do not show any evidence for (110) surface facets, so they should revise their description of the rods and correct this. Also, they circle a few of the bright dots in the image b and d, but too many of these dots have similar intensity. The numbers of bright dots should be consistent with the Pt loading, in terms of #of atoms/nm². The contrast from the Ce columns makes it difficult to properly identify the Pt atoms, hence it is best to look at the thin regions of the sample where the contrast from the support is a minimum. Since the AC-STEM image is a slice through the sample, and the Pt atoms are all at the surface, the image can be used to quantify the Pt loading. See ref [2] for a description of how the numbers of Pt single atoms per unit area can be consistent with bulk loading and surface analysis.

2) The loss of the CO band after flowing helium (Figure 2 d) cannot be used to infer weaker bonding of CO to Pt. As shown in ref 46, the lattice oxygen can react with the CO causing formation of CO₂ at ambient temperature. In that study, to check the strength of the CO binding with the Pt, the authors first depleted all interfacial oxygen by flowing CO during TPR. After that, the catalyst was cooled without exposing to any additional oxygen. It was found that the CO band could not be removed by flowing He. Only such a test could demonstrate the binding of CO to Pt, and whether that binding energy was altered by support doping. The experiment performed by the authors does not address this concern because the presence of active oxygen species at the Pt-ceria interface allows the strongly bound CO to be reacted easily at room temperature.

[1] S. Agarwal, L. Lefferts, B.L. Mojet, D.A.J.M. Ligthart, E.J.M. Hensen, D.R.G. Mitchell, W.J. Erasmus, B.G. Anderson, E.J. Olivier, J.H. Neethling, A.K. Datye, Exposed Surfaces on

Shape-Controlled Ceria Nanoparticles Revealed through AC-TEM and Water-Gas Shift Reactivity, *ChemSusChem*, 6 (2013) 1898-1906.

[2] D. Kunwar, S. Zhou, A. DeLaRiva, E.J. Peterson, H. Xiong, X.I. Pereira-Hernández, S.C. Purdy, R. ter Veen, H.H. Brongersma, J.T. Miller, H. Hashiguchi, L. Kovarik, S. Lin, H. Guo, Y. Wang, A.K. Datye, Stabilizing High Metal Loadings of Thermally Stable Platinum Single Atoms on an Industrial Catalyst Support, *ACS Catal.*, 9 (2019) 3978-3990.

Reviewer #3:

Remarks to the Author:

I did appreciate that authors greatly improved the quality of their manuscript, compared to initial submission. However, I still cannot suggest the possible publication of this work in *Nature Communications*.

1. Insights on catalytic mechanism are similar to previous studies, in which the sub-nanometer Pt clusters exhibit the best activity for CO oxidation.
2. Authors claimed the travel ban or something else on the in situ experiments. I still think it's worth to perform those experiments to reveal the dynamic changes of all species during the catalytic process.
3. In Figure R16, authors mis-understood my comments. Herein, I emphasized on the spatial distribution of Cu and Pt clusters instead of the distance between two Pt clusters. How make sure the contribution of Cu on Pt electronic structures., especially for such a low Cu level.
4. Contribution of Cu on the catalytic performance is still unclear. EDS cannot get the conclusion that Cu is enriched at Pt/CeO₂ interfaces.
5. I still do not trust the DFT calculations here, which were far away from the real catalysts.

Response to reviewers

Reviewers' comments:

Reviewer #1:

The authors have successful improved the manuscript, which in my opinion now can be published as it is.

Author reply:

We appreciate the reviewer's recognition on our work and thank him/her for help improving the draft. We believe our work will be of general interest to the catalysis community.

Reviewer #2:

The authors have revised the manuscript in response to the reviewer comments and have addressed most of the concerns. The work shows that the presence of Cu dopants allows the Pt samples with sub-nanometer clusters to achieve a high reactivity. When the reactivity is properly normalized to total Pt atoms, the performance is comparable to the best Pt catalysts, and the work implies that oxygen activation of the support was achieved by the Cu dopants. The work is suitable for publication. I have only two revisions to suggest.

Author reply:

We thank the reviewer for his/her constructive comments on our manuscript. After considering the two suggestions from reviewer, we have strengthened the analysis of TEM and *in-situ* DRIFTS results in the revised manuscript. Below is our detailed response to reviewer's suggestions.

I>> The authors need to check the indexing of ceria lattice planes in Figure 1. The (110) lattice plane is forbidden by symmetry rules, it is the (220) that is allowed. They need to check their calibrations and also the planes allowed by symmetry

considerations. Furthermore, it has been shown in the literature that ceria rods expose (111) and (100) surfaces [1]. The authors do not show any evidence for (110) surface facets, so they should revise their description of the rods and correct this. Also, they circle a few of the bright dots in the image b and d, but too many of these dots have similar intensity. The numbers of bright dots should be consistent with the Pt loading, in terms of #of atoms/nm². The contrast from the Ce columns makes it difficult to properly identify the Pt atoms, hence it is best to look at the thin regions of the sample where the contrast from the support is a minimum. Since the AC-STEM image is a slice through the sample, and the Pt atoms are all at the surface, the image can be used to quantify the Pt loading. See ref [2] for a description of how the numbers of Pt single atoms per unit area can be consistent with bulk loading and surface analysis.

[1] S. Agarwal, L. Lefferts, B.L. Mojet, D.A.J.M. Ligthart, E.J.M. Hensen, D.R.G. Mitchell, W.J. Erasmus, B.G. Anderson, E.J. Olivier, J.H. Neethling, A.K. Datye, Exposed Surfaces on Shape-Controlled Ceria Nanoparticles Revealed through AC-TEM and Water-Gas Shift Reactivity, *ChemSusChem*, 6 (2013) 1898-1906.

[2] D. Kunwar, S. Zhou, A. DeLaRiva, E.J. Peterson, H. Xiong, X.I. Pereira-Hernández, S.C. Purdy, R. ter Veen, H.H. Brongersma, J.T. Miller, H. Hashiguchi, L. Kovarik, S. Lin, H. Guo, Y. Wang, A.K. Datye, Stabilizing High Metal Loadings of Thermally Stable Platinum Single Atoms on an Industrial Catalyst Support, *ACS Catal.*, 9 (2019) 3978-3990.

Author reply:

We appreciate the reviewer's comment on our analysis of the TEM results. We agree with the reviewer that (110) and (100) lattice planes are forbidden for face-centered cubic crystal (such as CeO₂) due to symmetry rules, which also do not exist in the X-ray diffraction pattern as predicted by Bragg's law in Fig. R1. We have carefully checked calibrations for d-spacing measurements of lattice fringes in Fig. 1 and Supplementary Fig. 1. Despite the same atomic arrangement in parallel crystal planes, we have taken the reviewer's suggestion and revised miller indices of CeO₂ support's lattice fringes to (220) and (200), corresponding to d-spacing of 0.19 nm

and 0.28 nm, respectively.

Figure R1. X-ray diffraction pattern of CeO₂ nanorod support.

CeO₂ nanorod supports in this study are prepared by the same hydrothermal method as our previous study (*Catal. Sci. Technol.* 2017, 7, 4462). We have found that the CeO₂ nanorod after calcination at 500 °C exposes (220) and (200) facets, while CeO₂ nanorod after calcination at 700 °C exposes (111) facet, which agree with a number of previous studies (*J. Phys. Chem. B* 2005, 109, 24380; *J. Am. Chem. Soc.* 2012, 134, 20585; *ACS Catal.* 2016, 6, 2265). As shown in Fig. R2, we can clearly observe the (220) and (200) crystal planes in CeO₂ nanorod. Meanwhile, the CeO₂ (111) crystal plane can also be observed that exhibits an angle of 54.7 °C from the exposed (200) facet, agreeing well with the atomic model of CeO₂ nanorod. It is interesting to note that authors in Ref. [1] find that (111) surface being the prominent facet for CeO₂ nanorod with a similar synthesis procedure. It is also reported that the CeO₂ nanorod and octahedral exposed by (111) and (100) facets exhibit similar Fourier transform infrared spectra for surface hydroxyl groups. It is possible that discrepancies on the exposed facets of CeO₂ nanorod in previous studies may be attributed to surface reconstructions that can result in the transform of CeO₂ (110) to (111) (*Angew. Chem. Int. Ed.* 2017, 56, 375).

Figure R2. TEM images of as prepared CeO₂ nanorods. Side views of atomic model of CeO₂ nanorod are presented with different visual angles.

We thank the reviewer for providing us very helpful references about the consistency between TEM images and Pt mass loading, which is of significance for the characterization of single atom catalyst. According to the calculation method reported in Ref. [2], the numbers of Pt single atoms per unit area for Pt₁/Ce and Pt₁/CeCu are estimated to 0.24 Pt/nm² and 0.59 Pt/nm² based on the specific surface area of 79.4 m²/g as reported in our previous study (*Catal. Sci. Technol.* 2017, 7, 4462) and the mass loading of Pt used in experiments. Since the strong contrast from CeO₂ lattice makes it difficult to distinguish Pt atoms in Fig. 1 b and d, we have presented the HAADF-STEM images at the thin regions of Pt₁/Ce and Pt₁/CeCu in Fig. R3. The numbers of Pt single atoms per unit area are 0.16 Pt/nm² and 0.41 Pt/nm² based on the area size of 8 × 8 nm², which are consistent with calculated per unit area density of Pt₁/Ce and Pt₁/CeCu catalysts. We have added these results and discussions on the HAADF-STEM images of Pt single atoms in the revised supplementary information.

Figure R3. HAADF-STEM images of (a) Pt₁/Ce and (b) Pt₁/CeCu. Atomically dispersed Pt atoms are marked by white circles.

Modification:

1. Revised the miller indices of CeO₂ support. “...with exposed controllable facets of (110) and (100) ...” was revised to “... with exposed controllable facets of (220) and (200) ...”

(First paragraph in page 5 in text)

“... show CeO₂ (110) facet-supported Pt clusters ...” was revised to “... show CeO₂ (220) facet-supported Pt clusters ...”

(Second paragraph in page 6 in text)

2. Revised the miller indices in Fig. 1.

(Fig. 1 in page 5 in text)

3. Revised Supplementary Fig. 1 and its caption. “The confirmed crystal planes of (220) and (200) based on the measured d-spacing, as well as their included angle of 45° imply the main exposed facets are (220) and (200), ... The observed CeO₂ (111) crystal plane exhibits an angle of 54.7 °C from the exposed (200) facet, agreeing well with the atomic model of CeO₂ nanorod.”

(Supplementary Fig. 1)

4. Added the discussion of the numbers of Pt single atoms per unit area for Pt₁/Ce and Pt₁/CeCu. “The numbers of Pt atoms per unit area are also consistent with that calculated by the Pt mass loading of Pt₁/Ce and Pt₁/CeCu catalysts (Supplementary

Fig. 5).”

(Second paragraph in page 6 in text)

5. Revised Supplementary Fig. 5 and its caption. “*The numbers of Pt single atoms per unit area for Pt₁/Ce and Pt₁/CeCu are estimated to 0.24 Pt/nm² and 0.59 Pt/nm² based on ... are consistent with that calculated based on the Pt atoms in the area with size of 8 × 8 nm².*”

(Supplementary Fig. 5)

2>> The loss of the CO band after flowing helium (Figure 2 d) cannot be used to infer weaker bonding of CO to Pt. As shown in ref 46, the lattice oxygen can react with the CO causing formation of CO₂ at ambient temperature. In that study, to check the strength of the CO binding with the Pt, the authors first depleted all interfacial oxygen by flowing CO during TPR. After that, the catalyst was cooled without exposing to any additional oxygen. It was found that the CO band could not be removed by flowing He. Only such a test could demonstrate the binding of CO to Pt, and whether that binding energy was altered by support doping. The experiment performed by the authors does not address this concern because the presence of active oxygen species at the Pt-ceria interface allows the strongly bound CO to be reacted easily at room temperature.

Author reply:

We thank the reviewer for his/her comments on our discussion about the CO bonding strength on Pt clusters. To clarify the effect of Cu dopants on the bonding strength of CO on Pt clusters, we have performed the *in-situ* DRIFTS experiment according to the reference mentioned by the reviewer (*ChemCatChem* 2020, 12, 1726). Pt_n/CeCu catalyst is pretreated at 300 °C under 30 mL/min of 1% vol. CO balanced by Ar to deplete the active oxygen at the interfaces. After 10 min, the catalyst is cooled down to room temperature under CO flow. Then, the *in-situ* DRIFTS spectrum of CO adsorption of Pt_n/CeCu is collected as shown in Fig. R4. When the flow is switched from CO to Ar, we can still find the loss of adsorbed CO molecules, indicating

weakened CO adsorption on Pt clusters due to Cu dopants. We have added the corresponding results and discussion in the revised manuscript.

Figure R4. *In situ* DRIFTS spectra of CO adsorption of Pt_n/CeCu after the pretreatment at 300 °C under CO flow. After CO exposure, Ar flow is continued and the spectrum is recorded at 10 min.

Modification:

1. Added the discussion of the *in-situ* DRIFTS spectra of CO adsorption. “*In order to eliminate the effects of interfacial active oxygen, the in situ DRIFTS spectra of CO adsorption of Pt_n/CeCu after pretreatment under CO flow have been collected, which still show the loss of adsorbed CO molecules indicating weakened CO bonding strength on Pt clusters due to Cu dopants (Supplementary Fig. 20).*”

(First paragraph in page 10 in text)

2. Added Supplementary Fig. 20 about the results of *in situ* DRIFTS spectra.

(Supplementary Fig. 20)

Reviewer #3:

I did appreciate that authors greatly improved the quality of their manuscript, compared to initial submission. However, I still cannot suggest the possible publication of this work in Nature Communications.

Author reply:

We appreciate the reviewer for taking his/her time to evaluate our revised manuscript. We have seriously considered the reviewer's comments and made our best efforts to improve the work. We have emphasized the novelty of our reported strategy for the precise interface control, especially the role of Cu dopants on the enhanced catalytic activity. More realistic DFT calculations have been performed to study the CO oxidation at the interface of Pt₁₄/Ce and Pt₁₄/CeCu to strengthen the reliability of our proposed catalytic mechanism. We hope the revised manuscript will deem fit for the publication in *Nature Communications*. Below is our point by point response to reviewer's comments.

I>> Insights on catalytic mechanism are similar to previous studies, in which the sub-nanometer Pt clusters exhibit the best activity for CO oxidation.

Author reply:

We agree that many previous studies have reported that sub-nanometer Pt clusters exhibit enhanced catalytic activity compared with atomically dispersed Pt single atoms on CeO₂ supports (*ACS Catalysis* 2015, 5, 5164; *Angew. Chem. Int. Ed.* 2017, 56, 13078; *Nat. Commun.* 2019, 10:1358; *Nat. Commun.* 2019, 10:3808; *ACS Catal.* 2019, 9, 10626). Nevertheless, our main contribution is that a very low concentration Cu dopants are demonstrated to greatly enhance the low-temperature catalytic activity of CeO₂ supported Pt clusters, and the realization of Pt cluster preparation by a redox-coupled atomic layer deposition method. The origin of enhanced activity has been attributed to the weakened CO bonding strength at interfacial Pt atoms and the activated interfacial oxygen due to the formation of Cu-O-Ce sites. We believe our preparation strategy is generally applicable to the design of highly efficient noble metal catalysts and is of general interest to the broad catalysis community. We have strengthened the discussion on the role of Cu dopants in our revised manuscript.

2>> *Authors claimed the travel ban or something else on the in situ experiments. I still think it's worth to perform those experiments to reveal the dynamic changes of all species during the catalytic process.*

Author reply:

We thank the reviewer for encouraging us to perform the *in-situ* experiments to reveal the dynamic changes of catalysts. We are aware that the *in-situ* XAS experiments can to some extent reveal the changes of the local structure and chemical states of catalytic sites during the catalytic reaction, which are of great significance for understanding the catalytic mechanism (*ACS Catalysis* 2015, 5, 5164; *ACS Catal.* 2019, 9, 5752; *ACS Catal.* 2019, 9, 10626). Unfortunately, the *in-situ* XAS method is still out of our reach currently due to various constraints, and it will be interesting to explore the dynamic changes of local fine structure at the interface during catalytic reaction in future studies. Nonetheless, our conclusion about the effects of interfacial structure and reducibility on the enhanced catalytic activity of our catalysts remain solid. The interfacial Pt sites are favorable in binding oxygen during CO oxidation leading to the higher coordinate number of Pt-O, which can deactivate the Pt sites as the *in-situ* XAS experiments show in previous study (*ACS Catalysis* 2015, 5, 5164). In this work, our *ex-situ* XAS experiments exhibit a preliminary result that the relatively small coordinate number of Pt-O and low chemical valence state of supported Pt clusters are beneficial to the enhanced catalytic activity, which is also consistent with the previous study (*ACS Catalysis* 2015, 5, 5164).

3>> *In Figure R16, authors mis-understood my comments. Herein, I emphasized on the spatial distribution of Cu and Pt clusters instead of the distance between two Pt clusters. How make sure the contribution of Cu on Pt electronic structures., especially for such a low Cu level.*

Author reply:

We thank the reviewer's comment on the spatial distribution of Cu and Pt clusters.

Since Cu dopants are introduced with a low concentration during the hydrothermal synthesis procedure, Cu dopants mostly reside in the lattice of CeO₂ by replacing Ce atoms, which agrees well with the Cu K-edge XANES and H₂-TPR results. Pt clusters are anchored to the surface of Cu doped CeO₂ supports as shown in Fig. 1. Our *in-situ* DRFITS spectra, CO-TPD and H₂-TPR results indicate that Cu dopants can not only activate the interfacial oxygen, but also weaken the CO bonding strength at interfacial Pt atoms. This is in line with the differential charge densities analysis of Pt₅/Ce and Pt₅/CeCu in Fig. R5. Since the low-valence Cu dopant shares less electron density with surrounding oxygen atoms than Ce atoms, the bonding between interfacial Pt and oxygen is strengthened with an enhanced polarization, which leads a weakened adsorption of CO molecule at interfacial Pt atoms. We have added the results and discussion of differential charge density in the revised manuscript.

Figure R5. Differential charge density ($\Delta\rho$) and the corresponding contours in the plane crossing the interfacial Pt and oxygen atoms of (a) Pt₅/Ce and (b) Pt₅/CeCu.

Modification:

1. Added the discussion of differential charge density. “*The differential charge densities indicate that the bonding between interfacial Pt and oxygen is strengthened with an enhanced polarization due to the low-valence Cu dopant, which leads a weakened adsorption of CO molecule at interfacial Pt atoms (Supplementary Fig. 31).*”

(Second paragraph in page 13 in text)

2. Added the results of differential charge density in Supplementary Fig. 31.

(Supplementary Fig. 31)

4>> Contribution of Cu on the catalytic performance is still unclear. EDS cannot get the conclusion that Cu is enriched at Pt/CeO₂ interfaces.

Author reply:

We thank the reviewer's insightful question on the role of Cu dopants on the catalytic performance and agree that the EDS elemental maps alone are not enough to support the conclusion of Cu enrichment at the Pt/CeO₂ interface. EDS elemental maps (Fig. 1f) show that the concentrations of Cu dopants near the Pt clusters are higher than other regions. Our work also implies that Cu dopants can not only activate the interfacial oxygen, but also weaken the adsorption of CO molecule at interfacial Pt atoms. The formation of Cu-O-Ce species in the lattice of CeO₂ results in the shifting of reduction peak to low temperature as demonstrated by the H₂-TPR results. This is consistent with the XPS and Raman results of increased concentration of oxygen vacancies. Moreover, to eliminate the effect of interfacial active oxygen on the loss of CO adsorption signals in the *in-situ* DRIFTS spectra, we have performed the *in-situ* DRIFTS experiments of CO adsorption on Pt_n/CeCu after the pretreatment at 300 °C under CO flow. As shown in Fig. R6, we can find continued loss of CO adsorption signals after switching to Ar flow, implying that Cu dopants can weaken the CO adsorption on the supported Pt clusters. Combining all these experimental evidences, it can be concluded that Cu dopants at the interfaces of Pt/CeO₂ play key role in the enhanced catalytic activity. In the revised manuscript, we have strengthened our analysis about the EDS result and the discussion on the role of Cu dopants.

Figure R6. *In situ* DRIFTS spectra of CO adsorption of Pt_n/CeCu after the pretreatment at 300 °C under CO flow. After CO exposure, Ar flow is continued and the spectrum is recorded at 10 min.

Modification:

1. “These results indicated that Cu is enriched at the Pt/CeO₂ interfaces where it can directly tune the reducibility of the support near the interfaces and affect the activity of composite catalyst.” was revised to “Clearly, the concentrations of Cu dopants near the Pt clusters are higher than other regions, which is beneficial to tune the reducibility of the support near the interfaces and affect the activity of composite catalyst.”.

(First paragraph in page 7 in text)

2. Added the discussion of the *in-situ* DRIFTS spectra of CO adsorption. “In order to eliminate the effects of interfacial active oxygen, the *in situ* DRIFTS spectra of CO adsorption of Pt_n/CeCu after pretreatment under CO flow have been collected, which still show the loss of adsorbed CO molecules indicating weakened CO bonding strength on Pt clusters due to Cu dopants (Supplementary Fig. 20).”

(First paragraph in page 10 in text)

3. Added Supplementary Fig. 20 about the results of *in situ* DRIFTS spectra.

(Supplementary Fig. 20)

4. “... the coupling of Cu-O-Ce sites and Pt clusters leads to the activated interfacial

oxygen and moderate CO adsorption strength due to the electron transfer at interfaces ...” was revised to “... Cu dopants at the interfaces can not only activate the interfacial oxygen, but also weaken the adsorption of CO molecule at interfacial Pt atoms ...”

(First paragraph in page 14 in text)

5>> I still do not trust the DFT calculations here, which were far away from the real catalysts.

Author reply:

We understand the reviewer’s concern on DFT calculation models. While DFT calculation models of metal/oxide interfaces are usually smaller than the real catalysts considering limited computational resources, they do help reveal effects of interfacial structures on the catalytic activity. Similar models composed by CeO₂ slab and Pt clusters with several atoms (Pt₄ ~ Pt₈) have been utilized to investigate the catalytic mechanism of CO oxidation, water-shift gas reaction and purification of volatile organic compounds in previous studies, giving valuable insights into the catalytic mechanism (*Nat. Mater.* 2011, 10, 310; *ACS Catal.* 2016, 6, 6151; *ACS Catal.* 2014, 4, 2088; *J. Am. Chem. Soc.* 2012, 134, 8968; *ACS Catal.* 2016, 6, 418). In this work, our DFT calculation models are motivated and constructed based on the HAADF-STEM images and XAS characterizations. Furthermore, we have also calculated the energetic routes of CO oxidation at the interface of Pt₁₄/Ce and Pt₁₄/CeCu models with a larger Pt₁₄ cluster as shown in Fig. R7. Despite the subtle differences in calculated activation barrier energies of CO oxidation with differently sized clusters, they all support the hypothesis of weakened CO adsorption at the interface Pt. In the updated manuscript, we have added the results and discussion of the energetic routes of CO oxidation on Pt₁₄/Ce and Pt₁₄/CeCu.

Figure R7. Energetic routes of a CO molecule oxidized by the lattice oxygen at the interfaces of Pt₁₄/Ce and Pt₁₄/CeCu.

Modification:

1. “... the O_v formation energies and CO adsorption energies at the interfaces of oxide slabs supported Pt₅ and Pt₁₄ cluster with larger size have also been calculated” was revised to “... the energetic routes of CO oxidation at the interfaces of oxide slabs supported a larger Pt₁₄ cluster have also been calculated (Supplementary Fig. 32), which exhibit the similar results to supported Pt₅ clusters”.

(Second paragraph in page 13 in text)

2. Revised Supplementary Fig. 32 by adding the results of energetic routes of CO oxidation at the interfaces of Pt₁₄/Ce and Pt₁₄/CeCu.

(Supplementary Fig. 32)